# Knowledge Diversion for Efficient Morphology Control and Policy Transfer

**Fu Feng** [1 2]   **Ruixiao Shi** [1 2]   **Yucheng Xie** [1 2]   **Jianlu Shen** [1 2]   **Jing Wang** [1 2]   **Xin Geng** [1 2]

## Abstract

Universal morphology control aims to learn a universal policy that generalizes across heterogeneous robot morphologies, with Transformer-based controllers emerging as a dominant choice. However, such architectures incur substantial computational costs, resulting in high deployment overhead, and existing methods exhibit limited cross-task generalization, necessitating training from scratch for each new task. To this end, we propose DivMorph, a modular training paradigm that leverages knowledge diversion to learn *decomposable controllers*. DivMorph factorizes randomly initialized Transformer weights into *basic knowledge units* via SVD and employs dynamic soft gating, conditioned on task and morphology embeddings, to adaptively modulate these units into universal *learngenes* and morphology- and task-specific *tailors* during training, thereby achieving knowledge disentanglement. By selectively activating relevant components, DivMorph adaptively recomposes the controller, enabling efficient policy deployment and effective policy transfer to novel tasks. Extensive experiments demonstrate that DivMorph achieves state-of-the-art performance, improving sample efficiency for cross-task transfer by $3.3\times$ and reducing model size for single-agent deployment by $16.7\times$.

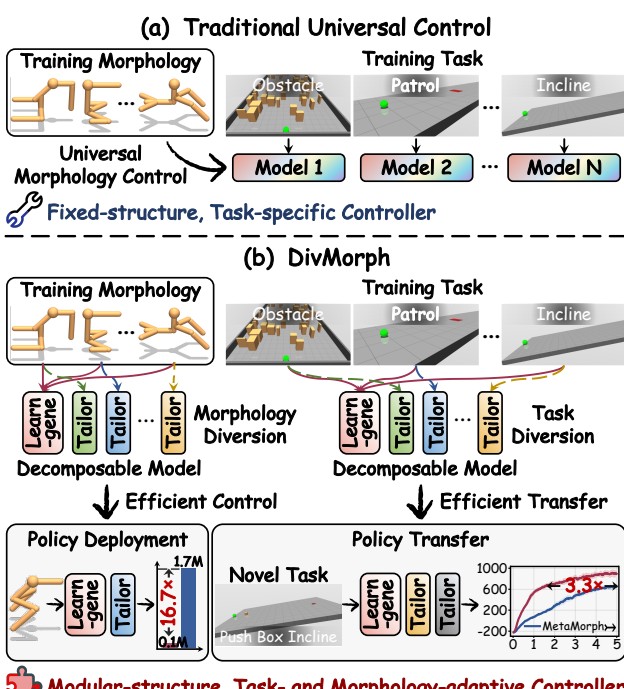

*Figure 1.* (a) Traditional universal morphology control adopts a fixed-structure controller shared across morphologies, yet remains task-specific and requires retraining for each task. (b) DivMorph leverages knowledge diversion to learn a modular, decomposable network that disentangles morphology- and task-specific components, enabling adaptive recomposition for universal morphology control while achieving $16.7\times$ smaller controllers for a given morphology and $3.3\times$ higher sample efficiency on novel tasks.

## 1. Introduction

Reinforcement learning (RL) is a fundamental paradigm for robotic control (Ju et al., 2022; Gu et al., 2024), yet learned policies often exhibit limited transferability across heterogeneous morphologies and tasks (Wang et al., 2023b;

[1]School of Computer Science and Engineering, Southeast University, Nanjing, China [2]Key Laboratory of New Generation Artificial Intelligence Technology and Its Interdisciplinary Applications (Southeast University), Ministry of Education, China. Correspondence to: Jing Wang <wangjing91@seu.edu.cn>, Xin Geng <xgeng@seu.edu.cn>.

*Proceedings of the $43^{rd}$ International Conference on Machine Learning*, Seoul, South Korea. PMLR 306, 2026. Copyright 2026 by the author(s).

He et al., 2024). Variations in agent morphology typically necessitate distinct control policies, while even a fixed morphology demands task-specific behavioral strategies (Yang et al., 2020; Trabucco et al., 2022). Consequently, standard RL frameworks require training separate policies from scratch for each morphology–task pair, with limited capacity for adaptive generalization (Singh et al., 2022).

To improve policy reuse, universal morphology control has been proposed to learn a single controller that generalizes across diverse agent morphologies (Wang et al., 2018; Kurin et al., 2020; Luo et al., 2025). Recent work, such as MetaMorph (Gupta et al., 2022), advances this paradigm by adopting a Morphology-Aware Transformer that models each agent as a sequence of limbs. ModuMorph (Xiong

et al., 2023) further exploits morphology context to improve control performance. However, Transformer-based controllers incur significant memory and computational overhead, making per-agent deployment costly (Hu et al., 2024). HyperDistill (Xiong et al., 2024) reduces this overhead by distilling Transformer knowledge into a compact MLP via a hypernetwork, but the resulting compression compromises generalization to unseen morphologies.

Despite these advances, existing approaches primarily generalize at the morphology level (Fig. 1a), resulting in limited policy reuse across tasks, while their rigid, memory-intensive Transformer-based controllers impose substantial deployment overhead. This motivates a key question: *Can we design a structurally adaptable controller that facilitates efficient universal control across diverse morphologies while enabling effective policy transfer across tasks?*

Recently, knowledge diversion (KDiv) (Xie et al., 2025) has emerged as a novel pretraining paradigm that factorizes network weights into task-agnostic *learngenes* and task-specific *tailors*, enhancing modular reuse and cross-task transfer, and has been widely adopted in generative models (Xie et al., 2026; Xu et al., 2025). Building on this principle, we introduce DivMorph, which brings knowledge diversion into universal morphology control (Fig. 1b). During training, DivMorph disentangles common (i.e., morphology- and task-agnostic) knowledge from morphology- and task-specific knowledge, and encapsulates them into *learngenes* and corresponding *tailors*, respectively. This enables DivMorph to achieve effective policy transfer to novel tasks by leveraging transferable *learngenes*, and efficient policy deployment to specific morphologies by discarding irrelevant *tailors*.

To enable knowledge disentanglement across morphologies and tasks in Transformer-based controllers, DivMorph first decomposes each randomly initialized weight matrix $W$ via Singular Value Decomposition (SVD), i.e., $W = U\Sigma V^{\mathsf{T}}$. Each singular vector pair $(u_i, v_i)$ from $U$ and $V$ is treated as *a basic knowledge unit* and assigned to one of three types: **universal** *learngenes*, **morphology-specific** *tailors*, or **task-specific** *tailors*. During knowledge diversion, the singular values $\Sigma$ are modulated by *morphology-* and *task-aware gates*, which regulate the recombination of basic knowledge units conditioned on task embeddings extracted from task instructions using a pretrained *text encoder*, and morphology embeddings derived from each agent's limb-joint configuration via a *morphology encoder*. To improve parameter sharing and generalization, DivMorph replaces conventional binary gating mechanisms (Xie et al., 2025; Xu et al., 2025) with dynamic soft gates that assign continuous activation coefficients to tailors in a mixture-of-experts style (Zhou et al., 2022), thereby facilitating modular reuse and zero-shot generalization to novel morphologies and tasks.

We conduct extensive experiments in the UNIMAL design space (Gupta et al., 2021), which comprises robots with 15–20 DoFs that are capable of acquiring locomotion and mobile manipulation skills in complex, stochastic environments. To evaluate generalization across morphologies and tasks, we construct 100 training and 100 novel morphologies, along with 5 training and 5 novel tasks, following (Gupta et al., 2021; 2022). Remarkably, DivMorph achieves state-of-the-art performance, achieving $3\times$ higher sample efficiency when transferring policies to novel tasks. Moreover, for policy deployment on a given morphology, it reduces model size by $17\times$, with complexity comparable to that of an MLP controller, demonstrating its effectiveness for scalable and flexible policy reuse.

Our contributions are as follows: 1) We propose DivMorph, the first modular training paradigm for universal morphology control, enabling flexible recomposition for efficient policy transfer and deployment. 2) We introduce a dynamic soft gating mechanism that effectively models tasks and morphologies, enabling enhanced universal control and robust zero-shot policy generalization. 3) We extend the UNIMAL benchmark to more challenging cross-task policy transfer settings. Extensive experiments demonstrate that DivMorph achieves state-of-the-art performance with substantially improved convergence speed and performance.

## 2. Related Work

### 2.1. Universal Morphology Control

Universal morphology control aims to learn *a single controller* that can operate agents with diverse morphologies, thereby enabling scalable control over large morphology spaces (Nagabandi et al., 2018; Pathak et al., 2019; Patel & Song, 2025). Early MLP-based controllers struggle to handle heterogeneous state and action spaces (Ghadirzadeh et al., 2021; Feng et al., 2023). Subsequent work adopts graph neural networks to capture morphological topology by enabling structured communication among neighboring actuators (Wang et al., 2018; Huang et al., 2020).

Recently, MetaMorph (Gupta et al., 2022) advances this line of work with a Morphology-Aware Transformer (Sec. 3.2), while ModuMorph (Xiong et al., 2023) further exploits morphology context to improve control performance; however, Transformer-based controllers incur substantial computational overhead. HyperDistill (Xiong et al., 2024) mitigates this cost by distilling Transformer policies into compact MLPs via a hypernetwork, but the resulting controllers exhibit limited generalization to novel morphologies.

More importantly, existing methods achieve only limited morphology-level generalization and remain task-specific, necessitating retraining for each new task. By contrast, DivMorph applies knowledge diversion during training to disentangle morphology- and task-specific components, en-

abling efficient policy deployment across morphologies and effective cross-task policy transfer.

## 2.2. Learngene and Knowledge Diversion

LEARNGENE (Feng et al., 2025a) is a biologically inspired knowledge transfer paradigm that encapsulates task-agnostic knowledge into modular neural units, termed learngenes, enabling efficient adaptation while substantially improving generalization and mitigating negative transfer. Early learngene-based approaches typically condense task-agnostic knowledge at the layer level. GRL (Feng et al., 2025a) employs evolutionary strategies, while Heur-LG (Wang et al., 2022b) and Auto-LG (Wang et al., 2023a) rely on heuristics or meta-learning to identify transferable layers. Other variants (Xia et al., 2024; Xie et al., 2024; Feng et al., 2025b; 2026) further impose structural constraints to regulate parameter sharing, thereby enhancing transferability. Although effective, these methods exhibit limited knowledge disentanglement (Shen et al., 2026), which restricts adaptability in multi-task and cross-domain scenarios.

Knowledge Diversion (KDiv) (Xie et al., 2025) advances this line of work by decomposing parameters into task-agnostic learngenes and task-specific tailors, facilitating modular reuse via gated routing (Xie et al., 2026; Shi et al., 2025). We extend this paradigm to universal morphology control by disentangling policy networks along both morphology and task dimensions, enabling unified morphology–task control while supporting efficient deployment and effective transfer to novel morphologies and tasks.

## 3. Preliminary

### 3.1. Problem Formulation

We study the problem of learning a *universal policy* for each of $T$ tasks, with each policy controlling $K$ robots of diverse morphologies. For a given task $\tau$, the control of robot $\kappa$ is modeled as a contextual Markov Decision Process (Hallak et al., 2015), i.e., $\mathcal{M}_\kappa^{(\tau)} = (\mathcal{S}_\kappa, \mathcal{A}_\kappa, \mathcal{C}_\kappa, T_\kappa, R_\tau)$, where $\mathcal{S}_\kappa$, $\mathcal{A}_\kappa$, $\mathcal{C}_\kappa$ and $T_\kappa$ denote the state space, action space, morphology context and transition function, while $R_\tau$ denote the task-specific reward, respectively.

We consider robots drawn from a modular design space, where each robot is structured as a tree of basic limb nodes with a shared node-level state and action space. For a robot with $N_\kappa$ limbs, the state and action spaces are defined as $\mathcal{S}_\kappa = \{S_{\kappa_i}\}_{i=1}^{N_\kappa}$ and $\mathcal{A}_\kappa = \{A_{\kappa_i}\}_{i=1}^{N_\kappa}$. The morphology context $\mathcal{C}_\kappa$ comprises node-specific attributes (e.g., limb size, mass, initial position) together with an adjacency matrix encoding the topology of the morphology tree.

For each task, let $s_\kappa^t$, $a_\kappa^t$, and $r_\kappa^t$ denote the state, action, and reward of robot $\kappa$ at time $t$, respectively, and let $c_\kappa$ denote

its morphology context. The objective is to learn a universal policy $\pi_\theta(a_\kappa^t \mid s_\kappa^t, c_\kappa)$ that maximizes the average return across all training morphologies: $\max_\theta \left[ \frac{1}{K} \sum_{\kappa=1}^{K} \sum_{t=0}^{H} r_\kappa^t \right]$, where $H$ denotes the task horizon. Beyond maximizing returns on the training morphologies and tasks, the universal policy $\pi_\theta$ is expected to efficiently control unseen morphologies $\kappa^* \notin \{1, \ldots, K\}$ and rapidly adapt to novel tasks $\tau^* \notin \{1, \ldots, T\}$, enabling zero-shot deployment and cross-task transfer with minimal additional training.

### 3.2. Morphology-Aware Transformer

Transformers can model interactions among elements in sets of arbitrary size, making them well suited for controlling robots with varying numbers of limbs. Leveraging this, MetaMorph (Gupta et al., 2022) introduces a Morphology-Aware Transformer for universal control of robots with heterogeneous morphologies (Fig. 2a).

At each time step $t$, limb $l$ of robot $\kappa$ receives the concatenation of its proprioceptive observation $s_{\kappa_l}^t$ and morphology context $c_{\kappa_l}$, which is projected into a node embedding $e_l$ via a shared linear layer. The set of embeddings $\{e_l\}_{l=1}^{N_\kappa}$ is processed by a Transformer encoder to capture inter-limb interactions, with each layer comprising a multi-head self-attention (MSA) followed by a feed-forward network (FFN).

In the MSA module, each attention head $A_i$ computes query, key, and value vectors from the node embeddings via learnable linear projections: $q_l^i = e_l W_q^i$, $k_l^i = e_l W_k^i$ and $v_l^i = e_l W_v^i$, where $W_q^i, W_k^i, W_v^i \in \mathbb{R}^{D \times d}$ are trainable projection matrices, $D$ is the embedding dimension, and $d$ is the head dimension. Self-attention is then performed as

$$A_i = \text{softmax}\left(\frac{Q_i K_i^\mathsf{T}}{\sqrt{d}}\right) V_i, \quad A_i \in \mathbb{R}^{N_\kappa \times d}, \qquad (1)$$

where $Q_i = [q_1^i, \ldots, q_{N_\kappa}^i]^\mathsf{T}$, $K_i = [k_1^i, \ldots, k_{N_\kappa}^i]^\mathsf{T}$, and $V_i = [v_1^i, \ldots, v_{N_\kappa}^i]^\mathsf{T}$. For a MSA module with $H$ heads, the outputs of all heads are concatenated and linearly projected through a learnable matrix $W_o \in \mathbb{R}^{Hd \times D}$:

$$\text{MSA} = \text{concat}(A_1, \ldots, A_h) W_o. \qquad (2)$$

The FFN comprises two linear transformations $W_{\text{in}} \in \mathbb{R}^{D \times D'}$ and $W_{\text{out}} \in \mathbb{R}^{D' \times D}$, with a GELU activation:

$$\text{FFN}(x) = \text{GELU}(x W_{\text{in}} + b_1) W_{\text{out}} + b_2, \qquad (3)$$

where $D'$ denotes the hidden layer dimension, and $b_1$ and $b_2$ are the corresponding biases.

The resulting per-node features are then fed into a shared decoder $W_{\text{dcd}}$ to generate per-node actions, which together form the robot's action vector at time $t$. When global exteroceptive observations (e.g., terrain height maps or goal information) are available, they are encoded by a dedicated MLP and concatenated with the per-node features prior to decoding to enrich the representation.

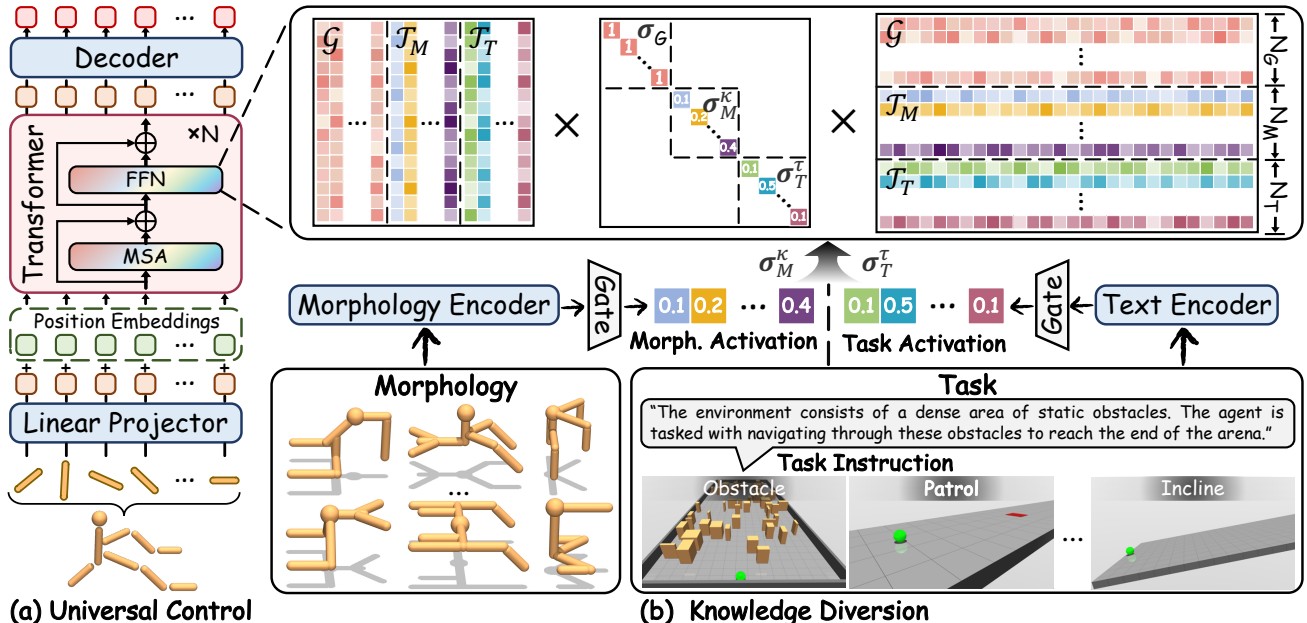

*Figure 2.* **Overview of DivMorph.** (a) Morphology-Aware Transformer. It represents each robot limb as a token and captures inter-limb interactions through a unified sequence representation, yielding a controller that generalizes across diverse morphologies. (b) Knowledge Diversion. Each randomly initialized weight matrix is factorized via SVD into basic knowledge units, subsequently partitioned into shared learngenes, morphology-specific tailors, and task-specific tailors. A dynamic soft gating mechanism selects the relevant tailors for each agent–task pair while jointly updating these basic units, enabling modular and disentangled representations across morphologies and tasks.

## 4. DivMorph

### 4.1. Knowledge Decomposition in Weight Matrices

To facilitate knowledge diversion for universal morphology control, we first decompose each weight matrix of the Morphology-Aware Transformer into a set of *basic knowledge units*, establishing a structured foundation for subsequent knowledge disentanglement and routing (Fig. 2).

Formally, let $\theta = \{W_q^{(1 \sim L)}, W_k^{(1 \sim L)}, W_v^{(1 \sim L)}, W_o^{(1 \sim L)}, W_{\text{in}}^{(1 \sim L)}, W_{\text{out}}^{(1 \sim L)}, W_{\text{dcd}}\}$[1] denote the collection of weight matrices across the $L$ Transformer layers. Here, $W_\star^{(l)}$ denotes the weight matrix of type $\star$ in layer $l$, with $\star \in \mathcal{S} = \{q, k, v, o, \text{in}, \text{out}, \text{dcd}\}$.

Following the decomposition strategy in (Xie et al., 2025; 2026), each matrix $W_\star^{(l)}$ is factorized via SVD:

$$W_\star^{(l)} = U_\star^{(l)} \Sigma_\star^{(l)} V_\star^{(l)\mathsf{T}} = \sum_{i=1}^{r} u_\star^{(l,i)} \sigma_\star^{(l,i)} v_\star^{(l,i)} \quad (4)$$

where $r$ is the rank of $W_\star^{(l)}$. Each rank-1 component $\Theta_\star^{(l,i)} = (u_\star^{(l,i)}, v_\star^{(l,i)})$ is treated as a *basic knowledge unit*.

During subsequent training, we directly update these basic units $\Theta_\star^{(l,i)}$ (instead of $W_\star^{(l)}$) to enable more fine-grained

---

[1] $W_q^{(1 \sim L)}$ denotes the set $\{W_q^{(1)}, \ldots, W_q^{(L)}\}$ for brevity. Similar notations throughout the paper follow this convention.

and modular adaptation. The full weight matrix $W_\star^{(l)}$ is then adaptively reconstructed, with each unit's contribution modulated by the singular values $\Sigma_\star^{(l)} = \{\sigma_\star^{(l,i)}\}_{i=1}^{r}$ under the control of a dynamic soft gating mechanism (Sec. 4.2).

### 4.2. Knowledge Diversion across Morphology and Task

To enable flexible adaptation across heterogeneous morphologies and tasks, the *basic knowledge units* are explicitly partitioned into *universal learngenes*

$$\mathcal{G}^{(l)} = [U_G^{(l)}, V_G^{(l)}] = \left\{ \Theta_\star^{(l,i)} \mid i \in [0, N_G), \star \in \mathcal{S} \right\}$$

and into *morphology-specific* or *task-specific tailors*

$$\mathcal{T}_M^{(l)} = [U_M^{(l)}, V_M^{(l)}] = \left\{ \Theta_\star^{(l,i)} \mid i \in [N_G, N_G + N_M), \star \in \mathcal{S} \right\}$$

$$\mathcal{T}_T^{(l)} = [U_T^{(l)}, V_T^{(l)}] = \left\{ \Theta_\star^{(l,i)} \mid i \in [r - N_T, r), \star \in \mathcal{S} \right\}$$

where the total number of components is $r = N_G + N_M + N_T$.

Unlike prior methods that employ binary gates conditioned on discrete classes (Xie et al., 2025) or condition labels (Xu et al., 2025), we introduce a **soft gating mechanism** that models similarities across morphologies and tasks, thereby enhancing generalization to unseen morphologies and tasks.

Specifically, for agent $\kappa$ performing task $\tau$, each task is paired with a textual instruction $I_\tau$ to enrich its semantics, which is encoded into a task embedding $e_\tau = E_T(I_\tau)$ via a

pretrained text encoder $E_T$ (Wang et al., 2022a). Similarly, the agent's morphology, specified by limb and joint parameters $P_\kappa$ (e.g., limb radius, density, joint type, and range), is encoded into a morphology embedding $e_\kappa = E_M(P_\kappa)$ via a dedicated morphology encoder $E_M$.

The task and morphology embeddings are fed into task- and morphology-aware gates, $G_T$ and $G_M$, producing soft weights over the corresponding tailor components:

$$\begin{aligned}
\boldsymbol{\sigma}_M^\kappa &= \mathrm{softmax}(\mathrm{TopK}(G_M(e_\kappa), k_M)) \in \mathbb{R}^{N_M}, \\
\boldsymbol{\sigma}_T^\tau &= \mathrm{softmax}(\mathrm{TopK}(G_T(e_\tau), k_T)) \in \mathbb{R}^{N_T}.
\end{aligned} \quad (5)$$

Here, $\mathrm{TopK}(\cdot, k)$ performs relevance-based filtering, retaining the $k$ largest gate scores prior to normalization. Learngenes are assigned unit scaling, i.e., $\boldsymbol{\sigma}_G = \mathbf{1}$. The complete set of singular values for a given agent–task pair is

$$\Sigma = [\Sigma_G, \Sigma_M^\kappa, \Sigma_T^\tau] = [\mathrm{diag}(\boldsymbol{\sigma}_G), \mathrm{diag}(\boldsymbol{\sigma}_M^\kappa), \mathrm{diag}(\boldsymbol{\sigma}_T^\tau)],$$

which specifies the relative contribution of each basic knowledge unit during reconstruction and is shared consistently across all layers ($1 \sim L$) and weight components ($\star \in \mathcal{S}$).

In this way, DivMorph achieves an explicit separation of morphology- and task-specific knowledge by jointly optimizing learngenes $\mathcal{G}^{(l)}$, tailors $\mathcal{T}_{\{M,T\}}^{(l)}$ and gates $G_{\{M,T\}}$ via **policy distillation**, which in turn indirectly updates the reconstructed weight matrix $W^{(l)}$ in Eq. (4) (see Alg. 1).

During knowledge diversion and agent training, we impose an orthogonality constraint on the square factor (either $U$ or $V$) via a Cayley transform (Trockman & Kolter, 2021), ensuring it remains on the orthogonal manifold (Lezcano Casado, 2019) and thereby stabilizing the factorization throughout RL training (see App. A for a brief proof).

### 4.3. Efficient Deployment and Task Adaptation

DivMorph trains a decomposable controller via knowledge diversion, enabling flexible composition across morphologies and tasks. Leveraging the semantic generalization capabilities of the text and morphology encoders, the morphology- and task-aware gates acquire transferable routing behaviors, allowing the controller to *zero-shot* adapt to unseen morphologies and semantically related tasks by selectively activating the corresponding tailors (Sec. 6.3.3).

Thus, for an agent $\kappa^*$ performing task $\tau^*$, whether its morphology $M_{\kappa*}$ and task instruction $I_{\tau*}$ are seen or novel, they are first encoded and fed into the morphology- and task-aware gates. The gates generate **sparse** soft routing weights $\boldsymbol{\sigma}_M^{\kappa^*}$ and $\boldsymbol{\sigma}_T^{\tau^*}$ via TopK selection (Eq. (5)). Exploiting this sparsity, **tailors not selected by the gates can be pruned**, enabling efficient deployment while retaining the functional

capacity of the controller, resulting in the effective weight:

$$\begin{aligned}
\widetilde{W}^{(l)} = U_G^{(l)} \Sigma_G V_G^{(l)\mathsf{T}} &+ \mathbf{1}_{[\boldsymbol{\sigma}_M^{\kappa^*} > 0]} \cdot (U_M^{(l)} \Sigma_M^{\kappa^*} V_M^{(l)\mathsf{T}}) \\
&+ \mathbf{1}_{[\boldsymbol{\sigma}_T^{\tau^*} > 0]} \cdot (U_T^{(l)} \Sigma_T^{\tau^*} V_T^{(l)\mathsf{T}})
\end{aligned} \quad (6)$$

where $\mathbf{1}_{[\cdot]}$ denotes an indicator function that selects the nonzero entries of the sparse routing vectors.

For tasks or morphologies with substantial shifts, the controller achieves *efficient* adaptation with minimal computational overhead by updating the selected tailors and applying only limited refinements to the learngenes, since the learngenes already encapsulate morphology- and task-agnostic knowledge, while the morphology- and task-aware gates are kept fixed to retain generalization (see App. C for details).

## 5. Experimental Setup

**Environments** We evaluate DivControl on the UNIMAL design space, employing 100 training and 100 novel morphologies as defined in MetaMorph (Gupta et al., 2022). We consider 10 tasks with diverse objectives and scenarios (Gupta et al., 2021), where Flat Terrain (FT), Variable Terrain (VT), Incline, Obstacle, and Patrol serve as training tasks for knowledge diversion, and Exploration, Escape, Point Navigation, Push Box Incline, and Manipulate Box as novel tasks for evaluation. Details are provided in App. D.

**Baselines** We compare DivMorph with representative methods for morphology-aware control and cross-task transfer: **1) MetaMorph** (Gupta et al., 2022) introduces the Morphology-Aware Transformer, the first Transformer-based controller enabling universal control across a modular robot design space. For a fair comparison, we additionally implement two policy-transfer variants of MetaMorph: **2) MetaMorph**↪ directly transfers the policy from the most similar training task to the novel task. **3) MetaMorph**⤳ distills policies from all training tasks into a single unified network for transfer. **4) ModuMorph** (Xiong et al., 2023) extends MetaMorph with morphology-conditioned fixed attention for improved morphology control. **5) HyperDistill** (Xiong et al., 2024) distills ModuMorph's knowledge into a hypernetwork that generates morphology-specific MLP controllers for efficient inference. **6) GRL** (Feng et al., 2025a) condenses task-agnostic knowledge into inheritable neural fragments via population-based evolution, enabling effective knowledge transfer across tasks. We adapt GRL to universal morphology control, enabling generalization across diverse robot morphologies.

**Ablations** We conduct three ablations to evaluate the individual contributions of DivMorph's key components. **1) w/o KDiv.** We separately ablate knowledge diversion in the Transformer blocks and decoder to assess their respective

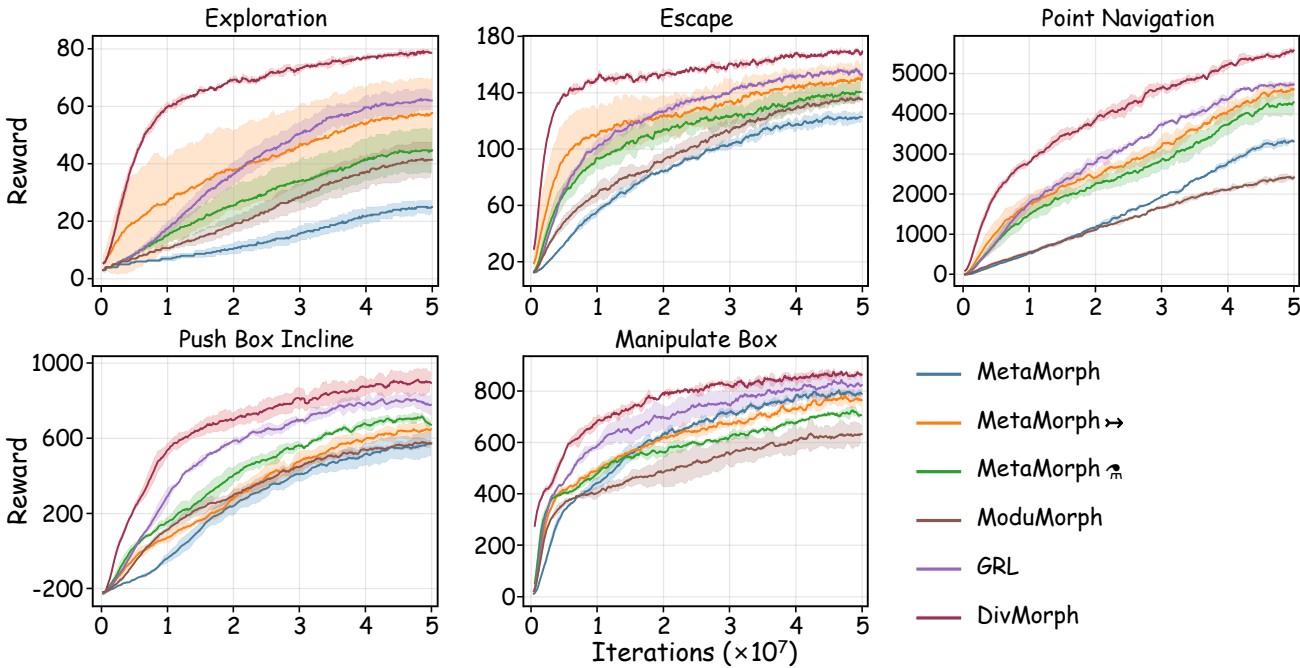

*Figure 3.* **Comparison of policy transfer performance on novel tasks with training morphologies.** Training curves show the mean and standard deviation of rewards for 100 UNIMAL robots with *training morphologies*, averaged over 3 runs per task. DivMorph consistently outperforms baselines, demonstrating higher sample efficiency across all tasks.

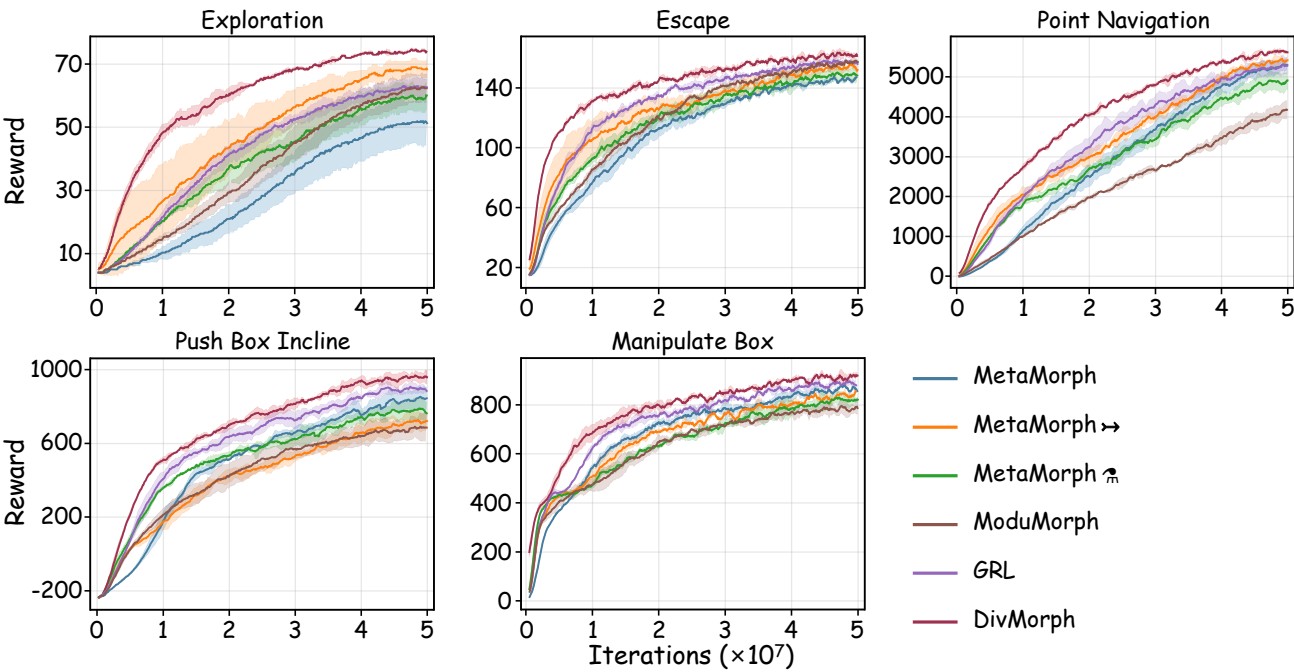

*Figure 4.* **Comparison of policy transfer performance on novel tasks with novel morphologies.** Training curves show the mean and standard deviation of rewards for 100 UNIMAL robots with *novel morphologies*, averaged over 3 runs per task. DivMorph still maintains a clear advantage, further demonstrating its robust generalization and rapid adaptation to unseen morphologies and tasks.

impact on policy adaptation across morphologies and tasks. **2) w/o Orthogonality.** We remove the orthogonality constraint on $U$ or $V$ to evaluate its impact on stabilizing RL training of the factorized parameters. **3) w/o Multi-task**

**Distillation.** The comparison between MetaMorph and MetaMorph⚓ isolates the effect of multi-task distillation on policy transfer and generalization. Detailed results and analysis can be found in Sec. 6.3

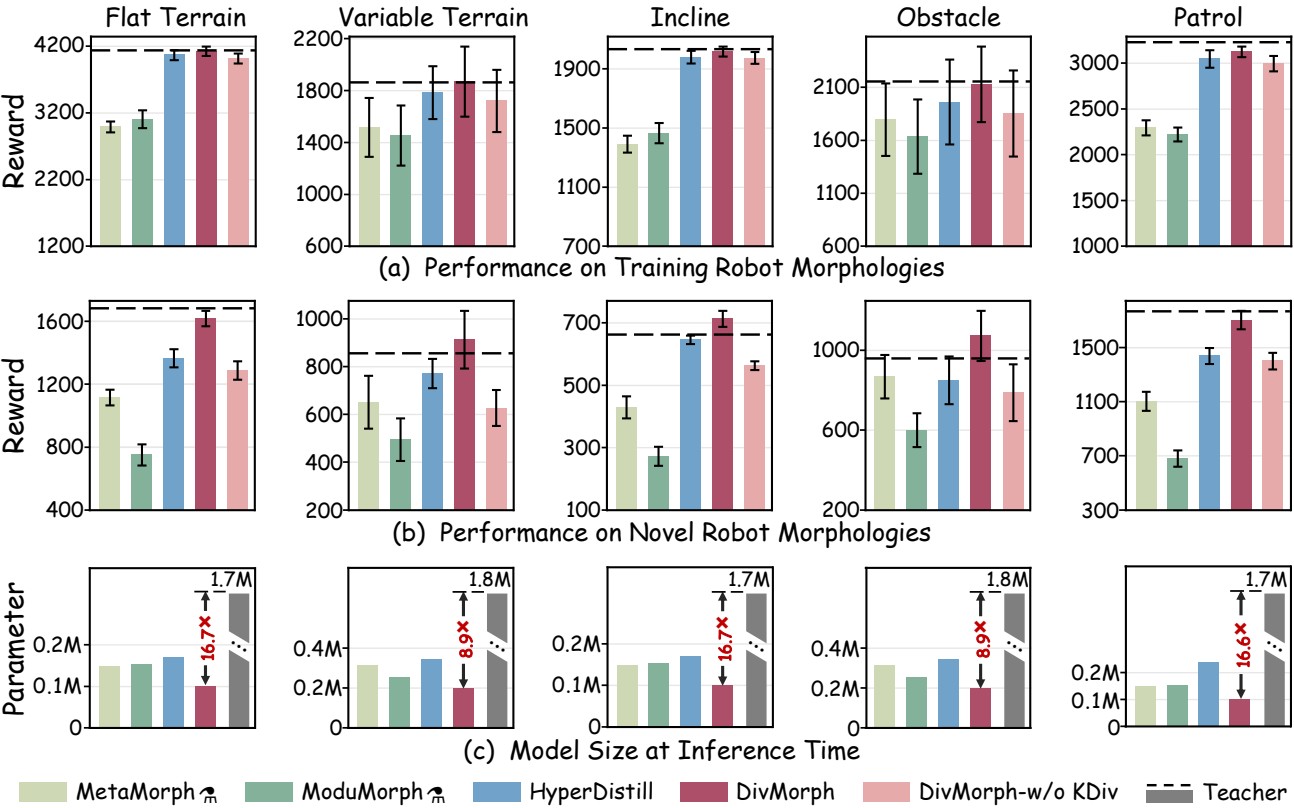

*Figure 5.* **Comparison of single-agent control performance on training and novel morphologies.** Bars indicate the mean and standard deviation of rewards across 100 UNIMAL robots for training (**a**) and novel morphologies (**b**), averaged over 3 runs per task, along with the corresponding model parameter counts (**c**). DivMorph achieves substantial model compression while matching the teacher models (i.e., MetaMorph pre-trained policies) on training morphologies, which approximate an upper performance bound. On novel morphologies, DivMorph demonstrates strong zero-shot generalization, even surpassing the teacher on several tasks (e.g., Incline and Obstacle).

## 6. Results

### 6.1. Effective Policy Transfer to Novel Tasks

**Universal Control of Training Morphologies on Novel Tasks.** DivMorph factorizes the Morphology-Aware Transformer into universal learngenes and morphology- and task-specific tailors via knowledge diversion. We first evaluate policy transfer to novel tasks under *universal morphology control* on training morphologies, thereby isolating the contribution of task-level knowledge diversion.

As shown in Fig. 3, DivMorph consistently outperforms all baselines, achieving faster adaptation and higher final returns. Compared with MetaMorph and ModuMorph, DivMorph improves sample efficiency by approximately $3\times$ to $15\times$, with particularly pronounced gains in the early training stages. Moreover, several tasks exhibit strong zero-shot performance, reflected in substantially higher initial returns, enabled by semantic generalization from task instructions.

Policy transfer generally outperforms training from scratch, but its effectiveness declines when transferred priors are excessive or misaligned with the target task. MetaMorph⚓

underperforms MetaMorph↦ because **indiscriminately aggregating** multiple source policies introduces redundant and task-inconsistent priors, which ultimately hinders generalization. Furthermore, both MetaMorph⚓ and MetaMorph↦ exhibit marked negative transfer on tasks with substantial behavioral mismatch (e.g., Manipulate Box), where inappropriate priors hinder efficient exploration.

In contrast, GRL achieves competitive performance by transferring only task-agnostic fragments, preserving flexibility for novel skills while avoiding irrelevant priors, thereby highlighting the necessity of DivMorph's knowledge diversion to isolate task-agnostic knowledge.

**Universal Control of Novel Morphologies on Novel Tasks.** We further evaluate joint policy transfer to novel morphologies and tasks (Fig. 4), where naively reusing full policies (i.e., MetaMorph⚓ and MetaMorph↦) leads to markedly stronger negative transfer and degraded performance. In contrast, DivMorph integrates task-agnostic learngenes with a task-aware gate that selectively routes the most compatible tailors for each target task, mitigating negative transfer while enabling flexible and effective policy adaptation. These re-

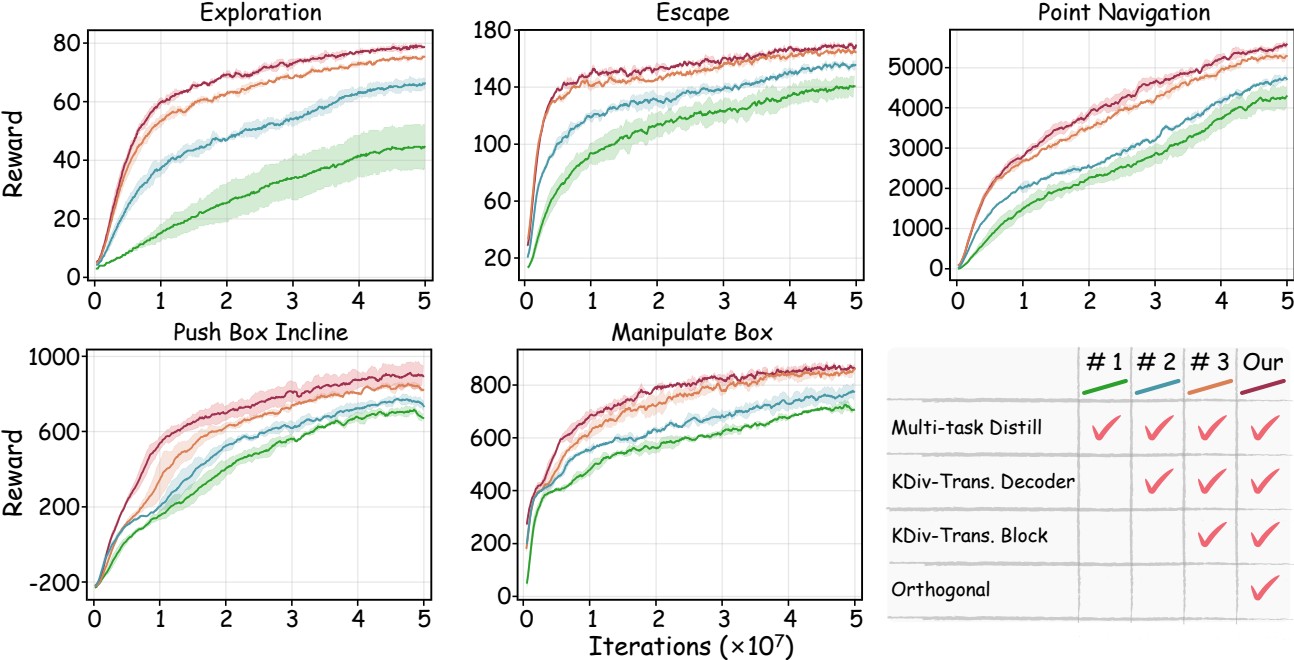

*Figure 6.* **Ablation study on DivMorph.** Training curves compare DivMorph with variants that remove block-level diversion, decoder-level diversion, or orthogonality constraints, illustrating their respective roles in learning efficiency, training stability, and final performance.

sults underscore the critical role of diverting task-agnostic knowledge and recombining task-specific knowledge for scalable cross-task generalization.

### 6.2. Efficient Deployment for Morphology Control

DivMorph disentangles task- and morphology-specific knowledge into independent learngenes and tailors, producing a modular controller that can be flexibly recombined at deployment. By selectively activating relevant components and *pruning irrelevant ones* (Eq. (6)), the controller adapts efficiently to diverse target morphologies and tasks while substantially reducing the deployed model size (Fig. 5).

For deployment on training morphologies, DivMorph attains performance comparable to the teacher—approaching an upper bound—while compressing the model by $8.9\times$ and $16.7\times$ (Fig. 5a,c). This highlights that, relative to conventional binary gates, the dynamic soft gate provides enhanced representational capacity, allowing decomposable Transformers to perform effective knowledge diversion while maintaining high-fidelity morphology encoding.

Importantly, the morphology encoder and morphology-aware gate jointly support robust cross-morphology generalization, allowing DivMorph to maintain superior *zero-shot performance* on unseen agents and, in several tasks, even exceed the transferred teacher policy (Fig. 5b). In contrast, HyperDistill achieves strong compression on training morphologies but generalizes poorly to novel ones; its hypernetwork, generating full MLP parameters, is prone to

overfitting, limiting adaptability under distribution shifts.

Ablations on morphology knowledge diversion (w/o KDiv) further emphasize the essential role of disentangling morphology-agnostic and morphology-specific components for efficient deployment and robust generalization.

### 6.3. Ablation and Analysis

#### 6.3.1. EFFECT OF KNOWLEDGE DIVERSION

Having preliminarily ablated morphology-level knowledge diversion in Sec. 6.2, we now perform a detailed ablation of task-level knowledge diversion. As shown in Fig. 6, diverting knowledge in both Transformer blocks and the decoder is critical for extracting task-agnostic representations; omitting either component (#2 and #3) impedes convergence and lowers final returns. Specifically, diversion at the Transformer block level captures generalizable task patterns, accelerating early learning, while decoder-level diversion encodes fine-grained adaptations, enhancing precise control and transfer, with both mechanisms together enabling scalable cross-task generalization.

#### 6.3.2. EFFECT OF ORTHOGONALITY CONSTRAINTS

We further ablate the orthogonality constraint (Fig. 6), showing that removing orthogonalization on $U$ or $V$ (each a square factor) degrades performance and destabilizes training. This stems from the fact that orthogonal matrices lie on a compact manifold, whereas non-orthogonal ones introduce unbounded and unstable degrees of freedom. Accordingly,

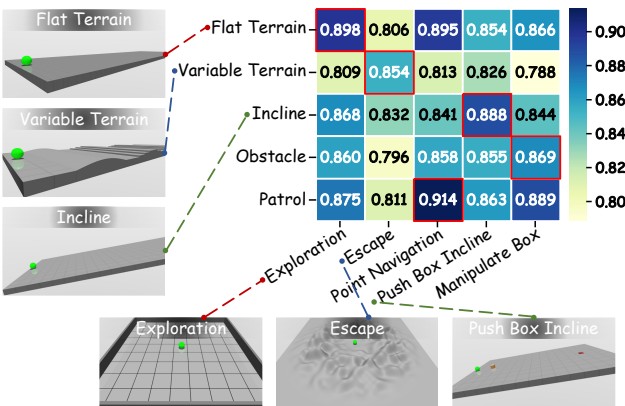

*Figure 7.* **Visualization of task-aware gate activations.** Correlation matrix of training and unseen tasks, illustrating inter-task similarity and task-specific routing patterns.

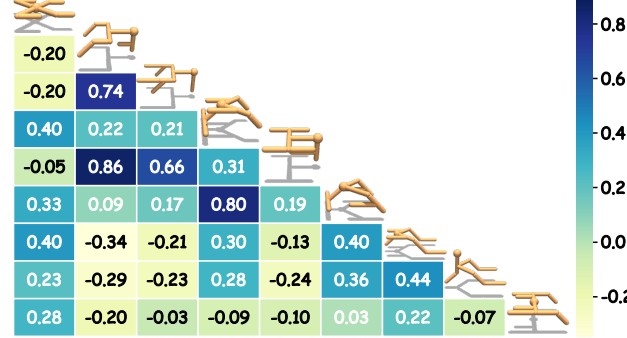

*Figure 8.* **Visualization of morphology-aware gate activations.** Correlation matrix of diverse robot morphologies, illustrating morphology-specific routing and captured generalization.

constraining the square factor during knowledge diversion regularizes the factorization process, preventing excessive and unstable degrees of freedom, thereby stabilizing RL training, and yielding stable and disentangled knowledge representations (a proof is given in App. A).

### 6.3.3. EFFECT OF MORPHOLOGY- & TASK-AWARE GATE

To better analyze DivMorph's adaptive recomposition mechanism, we visualize the task-aware and morphology-aware gates in Fig. 7 and Fig. 8, respectively. The task-aware gate exhibits a coherent semantic structure, wherein tasks with similar objectives or environmental contexts selectively activate highly overlapping subsets of tailors (e.g., Variable Terrain and Escape). This observation indicates that task instructions effectively complement and enrich the intrinsic task representations. Combined with the pre-trained text encoder, the gating mechanism provides a strong inductive bias that promotes robust cross-task generalization.

Similarly, combined with the morphology encoder, the morphology-aware gate encodes robot structures via topological and parametric features, generating consistent activations for structurally similar robots while distinguishing markedly different ones, thereby capturing shared regularities rather than memorizing individual designs and enabling reliable transfer across morphologies.

## 7. Conclusion

We introduce DivMorph, a novel framework for universal morphology control that performs knowledge diversion across morphology and tasks to construct a modular controller. By factorizing the Morphology-Aware Transformer into universal learngenes and morphology- and task-specific tailors, DivMorph enables flexible recomposition via selective activation of relevant components, facilitating efficient policy deployment and effective policy transfer to novel

tasks and morphologies. Extensive experiments demonstrate that DivMorph consistently outperforms existing methods, achieving notable gains in sample efficiency and overall performance while substantially reducing model size.

## 8. Limitations

Real-world deployment remains a fundamental limitation, as the diversity of simulated morphologies far exceeds the capabilities of existing robotic platforms, which are largely limited to standardized designs such as quadrupeds. Moreover, although Knowledge Diversion introduces additional computational overhead, it is performed entirely offline without extra environment interaction, making the cost negligible relative to the downstream acceleration gains.

## Impact Statement

The broader impact of DivMorph is the introduction of a modular reinforcement learning paradigm that enables the construction of decomposable controllers via knowledge diversion. By enabling systematic reuse and recomposition of transferable components, DivMorph promotes scalable policy learning and robust policy transfer across diverse morphologies and tasks, offering potential benefits for both theoretical research and real-world RL applications.

## Acknowledgement

We sincerely appreciate Freepik for contributing to the figure design. This research was supported by the Jiangsu Science Foundation (BG2024036, BK20243012), the National Natural Science Foundation of China (625B2045, 62125602, U24A20324, 92464301, 62306073), the New Cornerstone Science Foundation through the XPLORER PRIZE, the Fundamental Research Funds for the Central Universities (2242025K30024), and SEU Innovation Capability Enhancement Plan for Doctoral Students (CXJH_SEU 26023).

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

## A. Stability Analysis of Orthogonal Constraints in Knowledge Diversion

**Proposition.** Let $W \in \mathbb{R}^{m \times n}$ be a parameter matrix in a deep reinforcement learning controller, and consider its thin singular value decomposition

$$W = U \Sigma V^\mathsf{T},$$

where $U \in \mathbb{R}^{m \times r}$, $V \in \mathbb{R}^{n \times r}$, and diagonal $\Sigma \in \mathbb{R}^{r \times r}$, with $r = \min(m, n)$. When $W$ is constructed indirectly by optimizing the factors $(U, \Sigma, V)$, constraining at least one square factor (either $U$ or $V$) to lie on the orthogonal manifold during training prevents non-isometric scaling induced by unconstrained matrix multiplication.

As a consequence, gradient propagation with respect to the factorized parameters is well-conditioned, mitigating gradient explosion or vanishing along non-compact directions in parameter space, even when the other factor remains unconstrained.

**Observation 1 (Instability Induced by an Unconstrained Square Factor).** Consider a full-rank matrix $W \in \mathbb{R}^{m \times n}$ $(m < n)$ parameterized as

$$W = U \Sigma V^\mathsf{T},$$

where $U \in \mathbb{R}^{m \times m}$, $\Sigma \in \mathbb{R}^{m \times m}$, and $V \in \mathbb{R}^{n \times m}$ are all treated as trainable parameters.

To isolate the effect of the square factor $U$, define

$$Z := \Sigma V^\mathsf{T} \in \mathbb{R}^{m \times n},$$

so that the parameterization reduces to $W = UZ$.

When $U$ is unconstrained, gradient-based optimization of $(U, Z)$ may become unstable. In particular, non-isometric transformations induced by $U$ introduce directions along which gradients with respect to $Z$ can be arbitrarily amplified or attenuated, even when the induced update on $W$ remains bounded. This phenomenon reflects the presence of non-compact directions in the factorized parameter space, which may lead to gradient explosion or vanishing during training.

*Proof.* Since $U$ is a square matrix without structural constraints, it admits a singular value decomposition

$$U = ADB^\mathsf{T},$$

where $A, B \in \mathbb{R}^{m \times m}$ are orthogonal matrices and $D = \mathrm{diag}(\sigma_1, \dots, \sigma_m)$ contains the singular values of $U$, which are unrestricted and may vary over several orders of magnitude.

Substituting this decomposition into $W = UZ$ yields

$$W = AD(B^\mathsf{T} Z).$$

which decomposes the effect of $U$ into an isometric rotation ($A$ and $B^\mathsf{T}$) and an anisotropic scaling ($D$). While the orthogonal components preserve Euclidean norms, the diagonal scaling matrix $D$ can selectively amplify or suppress different directions in the representation $Z$.

For a loss function $\mathcal{L}$, the gradient with respect to $Z$ satisfies

$$\nabla_Z \mathcal{L} = U^\mathsf{T} \nabla_W \mathcal{L} = BDA^\mathsf{T} \nabla_W \mathcal{L}.$$

Taking norms on both sides gives

$$\|\nabla_Z \mathcal{L}\| \leqslant \|D\|_2 \|\nabla_W \mathcal{L}\| = \|U\|_2 \|\nabla_W \mathcal{L}\|.$$

Consequently, directions corresponding to large singular values of $U$ induce arbitrarily large gradient amplification, while those associated with small singular values lead to severe attenuation.

This imbalance persists even when the loss landscape with respect to $W$ is well-conditioned, indicating that the instability arises purely from the unconstrained factorization. Therefore, unconstrained updates to the square factor $U$ introduce non-compact directions in the parameter space, which fundamentally undermine the stability of gradient-based optimization.

□

**Observation 2 (Stability via an Orthogonal Square Factor).** Consider a full-rank matrix $W \in \mathbb{R}^{m \times n}$ parameterized as

$$W = UZ, \quad Z := \Sigma V^\mathsf{T},$$

where $U \in \mathbb{R}^{m \times m}$ is a learnable square factor, as in Observation 1. If $U$ is constrained to lie on the orthogonal manifold,

$$U^\mathsf{T} U = I,$$

then the mapping $Z \mapsto W = UZ$ is an isometry.

As a consequence, both the operator norm of $W$ and the gradient norm propagated to $Z$ are preserved under this reparameterization, yielding stable optimization behavior.

***Proof.*** Let $x \in \mathbb{R}^n$ be arbitrary. Since $Wx = U(Zx)$ and $U$ is orthogonal, left multiplication by $U$ preserves the Euclidean norm, implying

$$\|Wx\| = \|UZx\| = \|Zx\|.$$

Taking the supremum over all unit vectors $x$ yields

$$\|W\|_2 = \sup_{\|x\|=1} \|Wx\| = \sup_{\|x\|=1} \|Zx\| = \|Z\|_2,$$

Therefore, constraining $U$ to be orthogonal ensures that the spectral properties of $W$ are fully determined by $Z$, eliminating any anisotropic scaling that could otherwise arise from the square factor.

Now consider a differentiable loss function $\mathcal{L}$. By the chain rule applied to the reparameterization $W = UZ$, the gradient with respect to $Z$ is given by

$$\nabla_Z \mathcal{L} = U^\mathsf{T} \nabla_W \mathcal{L}.$$

Since orthogonal matrices preserve vector norms, i.e., $\|U^\mathsf{T} v\| = \|v\|$ for any $v$, it immediately that

$$\|\nabla_Z \mathcal{L}\| = \|\nabla_W \mathcal{L}\|,$$

Thus, gradient propagation through the orthogonal factor $U$ is norm-preserving, in sharp contrast to the unconstrained case described in Observation 1.

In summary, constraining the square factor $U$ to the orthogonal manifold renders the factorization $W = UZ$ an isometric reparameterization. This removes the non-compact directions responsible for instability in the unconstrained setting and ensures that optimizing the factors is as stable as directly optimizing $W$.

$\square$

**Proof of Proposition.** Consider a full-rank parameter matrix $W \in \mathbb{R}^{m \times n}$ with thin singular value decomposition

$$W = U\Sigma V^\mathsf{T}, \quad Z := \Sigma V^\mathsf{T}.$$

We first examine the unconstrained case. When no factor is restricted, the square factor $U \in \mathbb{R}^{m \times m}$ may admit arbitrarily large or small singular values. Writing its singular value decomposition as $U = ADB^\mathsf{T}$ reveals that left multiplication by $U$ induces anisotropic scaling of $Z$. As a consequence, gradient propagation obeys

$$\nabla_Z \mathcal{L} = U^\mathsf{T} \nabla_W \mathcal{L} \quad \Longrightarrow \quad \|\nabla_Z \mathcal{L}\| \leqslant \|U\|_2 \|\nabla_W \mathcal{L}\|,$$

indicating that unconstrained multiplicative factors introduce non-compact directions in parameter space. These directions may arbitrarily amplify or attenuate gradients, leading to potential instability during optimization.

We now turn to the orthogonally constrained setting. If the square factor $U$ is constrained to lie on the orthogonal manifold, $U^\mathsf{T} U = I$, then left multiplication by $U$ is an isometry. In this case, both the operator norm of the parameter matrix and the gradient norm are preserved:

$$\|W\|_2 = \|UZ\|_2 = \|Z\|_2, \quad \|\nabla_Z \mathcal{L}\| = \|\nabla_W \mathcal{L}\|.$$

Hence, the spectral properties of $W$ and the stability of gradient propagation are entirely determined by $Z$, and no unintended amplification or attenuation arises from the square factor.

Finally, recall that $Z = \Sigma V^{\mathsf{T}}$. Optimizing $\Sigma$ and $V$ independently corresponds to optimizing $Z$ through standard chain-rule propagation:

$$\nabla_\Sigma \mathcal{L} = \left(\frac{\partial Z}{\partial \Sigma}\right)^{\mathsf{T}} \nabla_Z \mathcal{L}, \quad \nabla_V \mathcal{L} = \left(\frac{\partial Z}{\partial V}\right)^{\mathsf{T}} \nabla_Z \mathcal{L}.$$

Since $\|\nabla_Z \mathcal{L}\|$ is norm-preserving with respect to $\|\nabla_W \mathcal{L}\|$ under orthogonal $U$, these factor-level gradients propagate stably to the parameter matrix $W$. Therefore, updating $(\Sigma, V)$ under an orthogonal square factor is as stable as directly updating $W$, while retaining the structural flexibility afforded by the SVD parameterization.

In summary, constraining the square factor $U$ to the orthogonal manifold removes the anisotropic scaling inherent to unconstrained multiplicative parameterizations. As a result, gradient-based optimization of $(U, \Sigma, V)$ enjoys the same stability guarantees as direct optimization of $W$, while enabling controlled, factor-wise updates aligned with the SVD structure.

$\square$

# B. Details on Knowledge Diversion and Policy Optimization

During the knowledge diversion phase, the policy network is optimized via policy distillation (Rusu et al., 2015), enabling the extraction of knowledge shared across different morphologies and tasks. Subsequently, for adaptation to novel tasks, the controller is fine-tuned through interactions with the environment using the Proximal Policy Optimization (PPO) algorithm (Schulman et al., 2017), allowing for task-specific performance refinement.

### B.1. Policy Distillation for Knowledge Diversion

Algorithm 1 provides the pseudo-code for the knowledge diversion mechanism in DivMorph, a procedure that systematically disentangles knowledge across both morphology and task dimensions.

Specifically, given a set of teacher policy networks trained on multiple tasks, we leverage their outputs to supervise the training of the student policy network $\pi_\theta$, whose parameters are factorized and constrained according to Eq. (4). The policy distillation loss is defined as

$$\mathcal{L}_{\text{distill}} = -\frac{1}{B} \sum_{i=1}^{B} \frac{\sum_{j=1}^{L}(1 - \text{mask}_{i,j}) \log \pi_\theta(a_{i,j}^{\text{teacher}} \mid s_i)}{\sum_{j=1}^{L}(1 - \text{mask}_{i,j})} \tag{7}$$

Here, $B$ denotes the batch size, $L$ the dimensionality of the action space, $a_i^{\text{teacher}}$ the action predicted by the teacher policy for input $s_i$, and $\text{mask}_i$ an observation mask indicating which elements of $s_i$ are ignored during distillation.

Minimizing $\mathcal{L}_{\text{distill}}$ allows the student network to imitate the ensemble behavior of multiple source-task policies, thereby explicitly disentangling morphology-specific and task-specific knowledge via the structured parameterization. This procedure yields a decomposable controller that encapsulates highly transferable knowledge while maintaining robust flexibility for adaptive composition according to target tasks. The composed controller can then be further fine-tuned for novel tasks and morphologies via PPO, as detailed in Section B.2.

### B.2. Policy Optimization via PPO Algorithm

Following the recomposition of the controller through selection of appropriate tailors according to the target task or agent morphology, the controller is further refined via Proximal Policy Optimization (PPO) (Schulman et al., 2017).

Specifically, given the initial actor policy $\pi_\theta(a|s)$ and critic value function $V_\phi(s)$ inherited from the knowledge diversion stage, PPO performs iterative updates using trajectories collected from interactions with the target environment. This procedure adapts the transferred policy to task- and morphology-specific dynamics while preserving the transferable knowledge captured during the diversion stage, ensuring both stability and sample-efficient learning.

The PPO algorithm optimizes two complementary objectives: the policy objective and the value objective. The policy objective maximizes a clipped surrogate function to constrain policy updates within a stable range, ensuring training stability by limiting overly large policy updates, which is defined as:

$$\mathcal{J}_\pi(\theta) = \mathbb{E}_t\left[\min\left(r_t(\theta)\hat{A}_t, \text{ clip}(r_t(\theta), 1 - \epsilon, 1 + \epsilon)\hat{A}_t\right)\right], \tag{8}$$

---

**Algorithm 1** Knowledge Diversion in Universal Morphology Control

---

**Input**: Actor network $\pi_\theta$ with parameters $\theta$; Teacher policy $\pi_{\text{teacher}}$; Observation set $\mathcal{D} = \{s^t_{\kappa_i,\tau_j}\}$, where each $s^t_{\kappa_i,\tau_j}$ is collected at time step $t$ under morphology $\kappa_i$ (parameters $P_{\kappa_i}$) and task $\tau_j$ (instruction $I_{\tau_j}$); Number of distillation epochs $N_{\text{ep}}$; Batch size $B$; Learning rate $\alpha$.

**Output**: Universal learngene $\mathcal{G}$, Morphology-specific tailor $\mathcal{T}_M$, Task-specific tailor $\mathcal{T}_T$

1: Randomly initialize parameters $\theta$ of $\pi_\theta$, and perform SVD on each weight matrix $W^{(l)}_\star$, obtaining the corresponding factors $U^{(l)}_\star$ and $V^{(l)}_\star$, which serve as the *basic knowledge units*

2: Enforce orthogonality by parameterizing a square factor via the Cayley transform. Taking $U$ as an example, we have

$$U = \text{Cayley}(A) := (I - A)^{-1}(I + A),$$

    where $A$ is a skew-symmetric matrix satisfying $A^\mathsf{T} = -A$.

3: **for** $ep = 1$ to $N_{\text{ep}}$ **do**

4:     **for** each mini-batch $\mathcal{B} = \{(s^t_{\kappa_i,\tau_j}, P_{\kappa_i}, I_{\tau_j})\}^B_{b=1}$ sampled from $\mathcal{D}$ **do**

5:         **for** each observation $(s^t_{\kappa_i,\tau_j}, P_{\kappa_i}, I_{\tau_j})$ **do**

6:             Encode morphology via morphology encoder $e_{\kappa_i} = E_M(P_{\kappa_i})$

7:             Encode task via *pre-trained* text encoder $e_{\tau_j} = E_T(I_{\tau_j})$

8:             Compute gating weights $\boldsymbol{\sigma}^{\kappa_i}_M$ and $\boldsymbol{\sigma}^{\tau_j}_T$ following Eq. (5)

9:             Forward propagate the actor to compute action $a_t$, where each weight matrix $W^{(l)}_\star$ in $\theta$ is constructed according to Eq. (4)

10:            Forward propagate the teacher policy to compute the reference action $a^{\text{teacher}}_t$

11:         **end for**

12:         Compute policy distillation loss $\mathcal{L}_{\text{distill}}$ according to Eq. (7)

13:         Backpropagate $\mathcal{L}_{\text{distill}}$ to compute gradients with respect to basic knowledge units $\nabla_U \mathcal{L}_{\text{distill}}, \nabla_V \mathcal{L}_{\text{distill}}$ and the gates and associated encoders $\nabla_{G_M} \mathcal{L}_{\text{distill}}, \nabla_{G_T} \mathcal{L}_{\text{distill}}, \nabla_{E_M} \mathcal{L}_{\text{distill}}$

14:         Update learngenes via gradient descent:

$$U^{(l)}_G \leftarrow U^{(l)}_G - \alpha \, \nabla_{U_G} \mathcal{L}_{\text{distill}}, \quad V^{(l)}_G \leftarrow V^{(l)}_G - \alpha \, \nabla_{V_G} \mathcal{L}_{\text{distill}}$$

15:         Update morphology tailors via gradient descent:

$$U^{(l)}_M \leftarrow U^{(l)}_M - \alpha \, \mathbf{1}_{[\boldsymbol{\sigma}^{\kappa_i}_M > 0]} \nabla_{U_M} \mathcal{L}_{\text{distill}}, \quad V^{(l)}_M \leftarrow V^{(l)}_M - \alpha \, \mathbf{1}_{[\boldsymbol{\sigma}^{\kappa_i}_M > 0]} \nabla_{V_M} \mathcal{L}_{\text{distill}}$$

16:         Update task tailors via gradient descent:

$$U^{(l)}_T \leftarrow U^{(l)}_T - \alpha \, \mathbf{1}_{[\boldsymbol{\sigma}^{\tau_j}_T > 0]} \nabla_{U_T} \mathcal{L}_{\text{distill}}, \quad V^{(l)}_T \leftarrow V^{(l)}_T - \alpha \, \mathbf{1}_{[\boldsymbol{\sigma}^{\tau_j}_T > 0]} \nabla_{V_T} \mathcal{L}_{\text{distill}}$$

17:         Update morphology- and task-aware gates and morphology encoders via gradient descent:

$$G_M \leftarrow G_M - \alpha \, \nabla_{G_M} \mathcal{L}_{\text{distill}}, \quad G_T \leftarrow G_T - \alpha \, \nabla_{G_T} \mathcal{L}_{\text{distill}}, \quad E_M \leftarrow E_M - \alpha \, \nabla_{E_M} \mathcal{L}_{\text{distill}}$$

18:     **end for**

19: **end for**

---

where $r_t(\theta) = \frac{\pi_\theta(a_t|s_t)}{\pi_{\theta_{\text{old}}}(a_t|s_t)}$ is the probability ratio between the updated and previous policy, $\hat{A}_t$ is the advantage estimated using Generalized Advantage Estimation (GAE), and $\epsilon$ is the clipping threshold. GAE computes the advantage as:

$$\hat{A}_t = \sum_{k=0}^{\infty} (\gamma\lambda)^k \delta_{t+k}, \quad \delta_t = r_t + \gamma V_\phi(s_{t+1}) - V_\phi(s_t) \tag{9}$$

where $\gamma$ is the discount factor and $\lambda$ balances bias and variance.

The value objective minimizes the mean squared error between the predicted value and empirical return:

$$\mathcal{J}_V(\phi) = \mathbb{E}_t\big[(V_\phi(s_t) - G_t)^2\big], \tag{10}$$

*Table 1.* Hyperparameters for effective policy transfer.

| Model Configuration | Value |
| --- | ---: |
| Number of layers | 5 |
| Number of attention heads | 2 |
| Ranks of learngene | 96 |
| Ranks of tailor | 32 |
| Activated tailor ranks | 16 |
| Embedding dimension | 128 |
| Feedforward dimension | 1024 |
| Decoder dimension | 128 |
| Non-linearity function | ReLU |
| Dropout | 0.1 |

| Knowledge Diversion | Value |
| --- | ---: |
| Optimizer | Adam |
| Scheduler | CosineAnnealingLR |
| Learning rate | 1e-3 |
| Epoch | 30 |
| Batch size | 5120 |

| PPO | Configuration |
| --- | ---: |
| Optimizer | Adam |
| Scheduler | CosineAnnealingLR |
| Initial learning rate | |
| -for *learngene* | 3e-5 |
| -for *others* | 3e-4 |
| Batch size | 5120 |
| Total timesteps | 5e7 |
| Discount $\gamma$ | 0.99 |
| GAE parameter $\lambda$ | 0.95 |
| PPO clipping parameter $\epsilon$ | 0.2 |
| Policy epochs | 8 |
| Entropy coefficient | 0.01 |
| Reward clipping | $[-10, 10]$ |
| Observation clipping | $[-10, 10]$ |
| Timesteps per rollout | 2560 |

with $G_t$ denoting the empirical return from time step $t$.

Since the learngene $\mathcal{G}$ encapsulates morphology- and task-agnostic knowledge with strong generalization, we assign it a relatively small learning rate to preserve policy stability. In contrast, the tailors $\mathcal{T}_M$ and $\mathcal{T}_T$, which encode morphology- and task- specific adaptations, are updated with a larger learning rate to enable efficient fine-tuning. This differential learning rate strategy balances stability and adaptability. Detailed hyperparameter settings are provided in Appendix C.

## C. Training Hyperparameters

Table 1 summarizes the key hyperparameters used to facilitate effective policy transfer to novel tasks. For baseline methods, we follow the hyperparameters reported in the original works (Gupta et al., 2022; Xiong et al., 2023; 2024; Feng et al., 2025a).

## D. Details of Training and Novel Tasks

We evaluate our approach on a set of 10 MuJoCo tasks (Todorov et al., 2012), specifically designed to comprehensively test the agent's capabilities across multiple dimensions. These tasks challenge the agent's agility, stability, and manipulation skills, by varying observations, objectives, and environmental interactions.

Collectively, these tasks constitute a diverse and challenging benchmark for systematically evaluating both the generalization and adaptability of the learned policies. We follow the experimental setup and task definitions provided in (Gupta et al., 2021; 2022). The corresponding task instructions, denoted by $I_\tau$, are defined as follows.

**Flat Terrain.** *"The environment consists of a flat arena. The agent is required to locomote across the surface and maximize its forward displacement over the course of an episode."*

**Variable Terrain.** *"The environment contains variable terrains, including hills, steps, and rubble. The agent must traverse these terrains and maximize its forward displacement over the course of an episode."*

**Incline.** *"The environment is a rectangular arena inclined at an angle of ten degrees. The agent is tasked with maintaining stable locomotion and maximizing forward displacement along the slope over the course of an episode."*

**Obstacle.** *"The environment consists of a dense area of static obstacles. The agent is tasked with navigating through these obstacles to reach the end of the arena."*

**Patrol.** *"The environment is a flat arena with two fixed goal locations. The agent is required to run back and forth between two goal locations."*

**Exploration.** *"The environment consists of a flat arena discretized into grid cells. The agent is required to explore the arena by locomoting and maximize the number of distinct cells visited over the course of an episode."*

**Escape.** *"The environment is a bowl-shaped terrain surrounded by small hills. The agent is initialized at the center and must traverse the hills to maximize its geodesic distance from the starting position."*

**Point Navigation.** *"The environment is a flat arena with randomly assigned goal locations. The agent starts at the center and must navigate to a randomly sampled goal location in each episode."*

**Push Box Incline.** *"The environment is an inclined plane containing a movable box. The agent is required to push the box along the incline to achieve maximal displacement toward the target direction."*

**Manipulate Box.** *"The environment consists of a flat arena containing a movable box. The agent is required to manipulate the box from a source location to a target location."*

