# OpenReview forum: "Knowledge Diversion for Efficient Morphology Control and Policy Transfer"
_ICML.cc/2026/Conference — ICML 2026 regular_

### Official Review · Reviewer_kjwa · 2026-02-25

**Soundness:** 2
**Presentation:** 3
**Significance:** 2
**Originality:** 3
**Overall Recommendation:** 4
**Confidence:** 4

**Summary:**

This paper proposes DivMorph targeted on solving universal morphology control problem with Transformer-based morphology-aware policies. The paper mainly propose two points on the policy arch:

1) SVD-factorize the metrics in blocks and decoder, and partitions them to shared learngenes/ morphology-specific tailors/ task-specific tailors. During update, only the learned units are updated;

2) Uses soft TopK gating conditioned on morphology embeddings and task-instruction embeddings to selectively activate the network, enable efficient deployment and stroger transfer ability.

Evaluation results on UNIMAL with transfer to novel tasks and novel morphologies, showing consistently strong performance over multiple baselines.

**Compliance With Llm Reviewing Policy:**

Affirmed.

**Final Justification:**

My main concerns lies in the herustic design for the N_X choices, but emprically they are not sensitive for tunning and behaves well across the reasonable parameter space. For the rest of the clarification problems, I think the rebuttal sufficiently solve my concerns and I would like to raise my score to 4 as my final recommendation.

**Key Questions For Authors:**

1) **Training of the morphology encoder**: ​
The paper introduces a dedicated morphology encoder, but the main text does not clearly specify how the encoder is trained. For example, what ground-truth/ objective it uses, and whether it is frozen during downstream PPO adaptation.

2) **Herustic SVD-component partitioning**:​
DivMorph hard-splits singular components into learngenes vs. morphology/task tailors by index, which is a completely herustic design. This may inject a strong inductive bias and raises practicality questions: how should N_G, N_M, N_T be selected for different task suites, and how sensitive is performance to these choices—especially since they introduce additional training-time parameters. I think some additional experiments on ablation with different N_G, N_M, N_T or analysis on the newly introduced parameters would help. I think this is one of the major concern/ question of the paper, since this point is the main contribution of the paper that requires more dedicated explanation and analysis.

3) **Training of the morphology encoder**:​
The paper introduces a dedicated morphology encoder, but the main text does not clearly specify how the encoder is trained. For example, what ground-truth/ objective it uses, and whether it is frozen during downstream PPO adaptation.

4) **Training details**:​
DivMorphs adds some nontrivial overheads (e.g. distillation stage, SVD factorization, Cayley orthogonalization, dual gates/encoders, TopK routing). The paper would benefit from reporting training-time details (wall-clock/GPU-hours, memory, caching/materialization strategy) and comparing overhead against baselines under matched compute. Since efficiency is a point the author want to highlight.

5) **Efficiency claim during deployment**:​
Figure 5 (and the abstract) shows compression relative to the teacher model size, which may not be a fair comparison for “efficiency,” since most other variants achieve the same level of parameters reduction.
Also, the reported reductions are primarily in parameter count. It remains unclear whether DivMorph can lead to measurable improvements in latency/ throughput relative to other efficiency-oriented variants/baselines,
Finally, just out of pure curiosty, I would like to know whether deployment cost is a real bottleneck in this setting since most evaluated cases seems are just state-based simulation.

6) **Benefit of gates in Sec. 6.3.3 not strongly evidence**:​
It is not obvious that gating is the primary driver of gains. It would be better to include more visualizations or specially designed experiments on the heurisitc designs.

I would raise my score if the concerns and questiones are well addressed!

**Limitations:**

The paper did not discuss its potential limitations in the final section. I would like to see how authors respond to the previous questions and being up front about the limitations of the work.

**Strengths And Weaknesses:**

**Strengths**:

1): Novel method: A clean way to bring knowledge-diversion style modularization into morphology-aware Transformer controllers via SVD + gated recomposition.

2): Strong, fairly extensive experiments: Evaluated on UNIMAL with transfer to novel tasks and novel morphologies, showing consistently strong performance over multiple baselines.

3): Clear ablations: Targeted component studies (e.g., block vs. decoder diversion, orthogonality constraints) that help attribute gains to key design choices.

**Weakness**:

I have several relevant questions regarding the specific tech details. See questiones raised below.

---

> ### Author Rebuttal · Authors · 2026-03-31
>
> Dear Reviewer kjwa,
>
> We sincerely appreciate your recognition of our innovation and contribution.
> Below, we provide our detailed response, with experimental tables and figures accessible via anonymous link as permitted by ICML26.
>
> **📎 Anonymous Link**\
> 👉 https://anonymous.4open.science/r/a-D81E/r.pdf
> >**Q1: Training of Morphology Encoder**
>
> Algorithm 1 details the training procedure of knowledge diversion.
> Specifically, **the Morphology Encoder (i.e., $E_M$) is jointly optimized** with the Gate, Learngene, and Tailor during distillation (line 17 in Alg.1), producing discriminative embeddings that enable the Morphology Gate to selectively activate the appropriate Tailor for each morphology.
>
> Once knowledge diversion (i.e., distillation) is complete, the embeddings for all training and test morphologies can be **precomputed and cached via a single forward pass through the resulting Morphology Encoder**, so that Morphology Encoder need not be invoked during subsequent PPO fine-tuning, incurring no additional computational overhead.
> >**Q2: Herustic SVD-component Partitioning**
>
> DivMorph is **generally insensitive** to the exact values of $N_G$, $N_T$ and $N_M$, **as long as extreme configurations are avoided (e.g., $N_G\gg N_T$)**, as the boundary between task-agnostic and task-specific knowledge is inherently diffuse.
>
> Specifically, for **task diversion**—with few tasks and an emphasis on cross-task knowledge reuse—we set $N_G$ slightly larger than $N_T$. In contrast, for **morphology diversion**—with many morphologies and a focus on efficient deployment—we set $N_G$ slightly smaller than $N_M$.
>
> We further conduct experiments on the choice of $N_G$, $N_T$ and $N_M$, with results for task diversion in Re_Fig.1 (see Reviewer iJDH(Q3)) and for morphology diversion in Re_Fig.2 (see Reviewer ho9M(Q3)) via the anonymous link.
> >**Q3: Same Question as Q1**
>
> >**Q4: Training Details**
>
> We focus on Universal Morphology Control, where a universal policy generalizes across heterogeneous robot morphologies.
> Training from scratch in this setting is costly **(~10 hours per task)** due to extensive environment interactions and Transformer overhead.
> Thus, we study **how to extract highly transferable knowledge from existing task policies to accelerate adaptation on new tasks**.
>
> The wall-clock overhead of our knowledge diversion stage is ≈0.5h, since it is performed offline without environment interactions.
> Re_Tab.1 breaks down each component's overhead:
> - The Gate, Encoder, and Router, are all single-layer MLPs, which incur negligible cost.
> - Orthogonalization adds minor overhead, as Cayley is applied **only to square matrices (≈10% of parameters)**, which are then **frozen during PPO fine-tuning** to ensure efficiency.
>
> Importantly, this 0.5h knowledge diversion overhead **yields substantial savings on new tasks**, and we have **already compared against matched-compute baselines** (e.g., MetaMorph⚗, which also uses multi-task distillation for policy transfer; see Sec.5).
> As shown in Re_Fig.4b, our method reaches MetaMorph⚗’s peak performance with minimal iterations, saving over 108h cumulatively across five new tasks [108h = (9+9+5+6+7)h × 3 seeds].
> The advantage is even greater compared to MetaMorph trained from scratch (see Re_Fig.4c), yielding cumulative savings of over 117h [(9+9+7+8+6)h × 3 seeds].
> >**Q5: Efficiency Claim during Deployment**
>
> As noted in HyperDistill[1], although a shared Transformer backbone yields more generalizable policies for agents with diverse morphologies than separate MLPs, its higher inference latency limits practical single-agent deployment.
> We have compared with HyperDistill in paper(Fig.5), **an efficiency-oriented baseline** that distills the unified policy into lightweight MLPs for efficient deployment.
>
> As suggested, we further report latency and throughput, and include the computational cost of a single-hidden-layer MLP, which we consider **a practical efficiency bound** for this setting (see Re_Tab.2 in anonymous link).
>
> [1] Distilling Morphology-Conditioned Hypernetworks for Efficient Universal Morphology Control, ICML'24
> >**Q6: Benefit of Gates**
>
> We have conducted the corresponding ablation study (Fig. 6 in paper), where **w/o KDiv corresponds to removing the gate**, as the gate is essential for knowledge diversion and its removal disables this process.
> Sec. 6.3.3 further illustrates its effectiveness by visualizing task- and morphology-level similarities captured by the gate.
>
> To further clarify the benefit of gate, we additionally analyze the top-K sampling mechanism (Re_Fig.3; see Reviewer iJDH(Q4)) and further conduct an ablation by replace dynamic soft gating with a discrete 0–1 mechanism (see Re_Fig.5; Reviewer pu2W(Q3)).
> >**Limitation**
>
> We will acknowledge the limitations of sim-to-real evaluation (see Reviewer ho9M(Q4)) and highlight the trade-off between the additional computation incurred by knowledge diversion and the substantial time savings achieved on new tasks.

---

> > ### Author Rebuttal · Reviewer_kjwa · 2026-04-01
> >
> > Thank you for the detailed replies and the additional experiments, which solve much of my concrens and questions! I would raise my score to 4.

---

> > > ### Author Response · Authors · 2026-04-03
> > >
> > > Dear Reviewer kjwa,
> > >
> > > We sincerely appreciate your thoughtful evaluation and are delighted that our rebuttal has addressed your concerns 😊.
> > >
> > > Best regards,
> > >
> > > Authors

---

### Official Review · Reviewer_iJDH · 2026-03-04

**Soundness:** 2
**Presentation:** 3
**Significance:** 4
**Originality:** 3
**Overall Recommendation:** 4
**Confidence:** 4

**Summary:**

This paper proposes DivMorph, a modular training paradigm for universal morphology control that addresses two limitations of existing Transformer-based controllers: high deployment cost and poor cross-task transferability. The core idea is to apply knowledge diversion to decompose Transformer weights via SVD into universal learngenes (task- and morphology-agnostic), morphology-specific tailors, and task-specific tailors. A dynamic soft gating mechanism, conditioned on task instructions (via a pretrained text encoder) and morphology parameters (via a dedicated encoder), produces continuous activation coefficients to route relevant tailors for each agent-task pair. Orthogonality constraints via the Cayley transform are imposed to stabilize training. At deployment, irrelevant tailors are pruned for model compression; for cross-task transfer, learngenes are retained and recombined with newly routed tailors followed by PPO fine-tuning. Experiments on 200 morphologies and 10 tasks in the UNIMAL design space show approximately 3-15x sample efficiency gains for novel-task transfer and up to 16.7x model compression, outperforming MetaMorph, ModuMorph, HyperDistill, and GRL baselines.

**Compliance With Llm Reviewing Policy:**

Affirmed.

**Final Justification:**

I appreciate the authors' detailed reply and additional analyses, which have solved most of my concerns. I am happy to keep my original evaluation (weak accept).

**Key Questions For Authors:**

1. Could the SVD-factorized architecture with soft gating be trained end-to-end via online multi-task RL, bypassing the distillation stage? If not, what specific challenges prevent this?
2. The rank allocation and TopK threshold are fixed throughout all experiments without justification or sensitivity analysis. What guided these choices, and how sensitive is performance to different rank ratios or sparsity levels?

**Limitations:**

The authors are encouraged to more thoroughly discuss the potential limitations of DivMorph in Section 7 or a dedicated Limitations section.

**Strengths And Weaknesses:**

**Strengths:**
1. **Well-motivated problem with a clear narrative.** The paper provides a coherent survey from monolithic controllers to universal morphology control and various morphology context exploitation strategies, clearly identifying two remaining bottlenecks — limited cross-task transferability and high deployment cost — and addressing both within a unified framework. The TopK + softmax soft gating further captures continuous similarities across morphologies and tasks, enhancing generalization to unseen agents and tasks.
2. **Solid theoretical justification for the orthogonality constraint.** Appendix A rigorously demonstrates that constraining the square SVD factor to the orthogonal manifold ensures norm-preserving gradient propagation, preventing gradient explosion/vanishing in the factorized parameterization. This provides meaningful theoretical grounding for a design choice critical to RL training stability.
3. **Comprehensive experimental evaluation.** The experiments systematically cover policy transfer (training/novel morphologies × novel tasks), single-agent deployment efficiency, ablation studies (knowledge diversion, orthogonality, multi-task distillation), and intuitive gate activation visualizations. This multi-faceted evaluation strengthens the empirical conclusions.
4. **Clear writing and presentation.** The paper is well written and easy to follow, with clean mathematical formulation and highly informative figures (especially Fig. 2 and Figs. 7-8).

**Weaknesses:**
1. Knowledge diversion requires fully trained teacher policies per task as a prerequisite, yet the cost of training these teachers (and subsequent distillation) seems to be excluded from the reported sample efficiency comparisons. Moreover, the paper does not clearly motivate why distillation is necessary rather than performing knowledge diversion directly during online RL training.
2. DivMorph leverages a pretrained text encoder for task embeddings, introducing substantial semantic inductive bias unavailable to any baseline. A natural alternative is simply concatenating task instruction embeddings with global exteroceptive observations as MetaMorph input, requiring no architectural change. Without analyzing such an alternative, it is difficult to disentangle how much improvement stems from knowledge diversion itself versus the pretrained language model's semantic prior.
3. The paper does not discuss any limitations of the proposed method, such as the dependency on pre-trained teacher policies and knowledge distillation.

---

> ### Author Rebuttal · Authors · 2026-03-31
>
> Dear Reviewer iJDH,
>
> We sincerely thank you for recognizing our motivation and theoretical foundations.
> Below, we provide our detailed response, with experimental tables and figures accessible via anonymous link as permitted by ICML26.
>
> **📎 Anonymous Link**\
> 👉 https://anonymous.4open.science/r/a-D81E/r.pdf
> >**Q1:Cost of Teachers and Online multi-task RL for Knowledge Diversion**
>
> Although training teachers is time-consuming, **it does not constitute a necessary overhead in our setting**.
> Our work builds on the Universal Morphology Control task and UNIMAL benchmark, allowing us to **leverage publicly available pretrained weights from other methods**.
>
> Moreover, **pretrained models are not intrinsic to our method, as nearly all baselines in our setting leverage them for fine-tuning**.
> For example, MetaMorph⚗ also distills knowledge from all pretrained teachers, while MetaMorph↣ transfers the teacher from the most similar training task for each new task.
>
> Consequently, the **sole additional overhead of our method is the knowledge diversion stage**, with a wall-clock cost of ≈0.5h, as it is performed offline without environment interactions (Re_Tab.1 in link details each component’s overhead).
>
> Importantly, **this overhead (≈0.5h) is negligible compared to the time saved on new tasks**: as shown in Re_Fig.4b, our method reaches MetaMorph⚗’s peak performance with minimal iterations, yielding over 108h of cumulative savings across five new tasks [(9+9+5+6+7)h × 3 seeds], with even greater savings compared to training from scratch (Re_Fig.4c).
>
> ---
>
> Based on the above, we adopt knowledge distillation rather than online RL, as **off-the-shelf pretrained models are readily available**, avoiding costly environment interactions.
>
> Nonetheless, our method can also **leverage online RL for knowledge diversion** when pretrained models are unavailable.
> As suggested, we supplement experiments on online multi-task RL for knowledge diversion (see Re_Fig.6).
> Compared to distillation, online learning **achieves similar transfer performance** on new tasks but incurs higher costs (~10h) due to environment interactions.
> >**Q2:Semantic Inductive Bias**
>
> As noted in Q1, **our baselines (e.g., MetaMorph↣) also perform knowledge transfer and benefit from semantic inductive bias**; for example, MetaMorph↣ transfers the teacher from the most similar training task, with **task similarity computed from task embeddings**.
>
> Essentially, this semantic inductive bias **does not directly encode agent-action semantics**.
> As suggested, we compare with a variant that concatenates task embeddings with global exteroceptive observations (see Re_Fig.7). The results show no significant difference from MetaMorph↣, indicating that **our method’s gains arise from knowledge diversion itself** rather than the pretrained language model’s semantic prior.
> >**Q3:Rank Allocation**
>
> In our experiments, each agent limb is encoded as a 128-dimensional embedding, setting the rank of the Transformer controller’s parameter matrices to 128.
> Accordingly, the Learngene($N_G$) and Tailor($N_T$) ranks are chosen such that $N_G+N_T=128$.
>
> For task diversion, we set $N_G=96$ and $N_T=32$, as the Learngene captures task-agnostic knowledge across 100 robot morphologies, while the Tailor, analogous to low-rank adaptation modules (e.g., LoRA), encodes task-specific information, hence $N_G>N_T$.
> As the boundary between task-agnostic and task-specific knowledge is inherently diffuse, **performance is generally robust to rank allocation**, except under extremely imbalanced settings.
>
> Further ablations in Re_Fig.1 show that a too-small Learngene rank (e.g., $N_G=32$, $N_T=96$) creates an information bottleneck, limiting general control priors, while an excessively large rank (e.g., $N_G=112$, $N_T=16$) restricts the Tailor’s capacity, hindering task-specific adaptation.
> We therefore adopt a 3:1 ratio to **balance generalization and specialization, with performance largely insensitive to moderate variations**.
>
> For morphology diversion, which prioritizes efficient deployment, we set $N_G$ slightly below $N_M$ (see Re_Fig.2 and Reviewer ho9M(Q3)).
> >**Q4:TopK**
>
> The number of active tailors per task (Top-K) is generally robust except at extreme values, as shown in Re_Fig.3.
> - Activating too few tailors limits task-specific expressiveness, forcing the Learngene to absorb excessive task-specific information and weakening generalization.
> - Activating too many tailors increases overlap across tasks, reducing routing sparsity and discrimination, and hindering task-specific feature capture.
>
> We therefore choose an intermediate value (TopK=16 for 32 Tailors), **observing similar performance for Top-K values around 16**, further demonstrating the robustness of our method.
> >**Limitation**
>
> We will dedicate a section to discuss potential limitations, such as sim-to-real evaluation and the trade-off between the extra computation introduced by knowledge diversion and the resulting time savings.

---

> > ### Author Rebuttal · Reviewer_iJDH · 2026-04-04
> >
> > I appreciate the authors' detailed reply and am happy to maintain my assessment.

---

> > > ### Author Response · Authors · 2026-04-04
> > >
> > > Dear Reviewer iJDH,
> > >
> > > We are glad that our rebuttal addressed your concerns, and we are grateful for your thoughtful assessment 😊.
> > >
> > > Best regards,
> > >
> > > Authors

---

### Official Review · Reviewer_pu2W · 2026-03-13

**Soundness:** 2
**Presentation:** 2
**Significance:** 2
**Originality:** 2
**Overall Recommendation:** 4
**Confidence:** 3

**Summary:**

This work proposes DivMorph, a modular training paradigm for learning cross-morphology control policies. The proposed method uses a 'knowledge diversion' idea and factorizes randomly initialized Transformer weights into knowledge units conditioned on task and morphology embeddings, in order to modulate these units into "learngenes" and "tailors". This factorization aims to achieve knowledge disentanglement and adaptively recomposes the controller for different robot morphologies, then it effectively transfers to novel tasks.

**Compliance With Llm Reviewing Policy:**

Affirmed.

**Key Questions For Authors:**

1. In figures 4 and 5, what is the qualitative behavioral difference between different rewards? Because the scale is in the thousands, it's hard to tell how differently the policies act for a e.g. 4000 reward vs 3500 reward.

2. What is the simulation engine and roughly what does the wall-clock time look like for each policy training run?

3. What is really the role of text description of each task? Besides the fact that it needs a pre-trained text encoder and potentially produces meaningful task embeddings, it does not seem necessary for the limited number of task variations studied here?

**Limitations:**

yes

**Strengths And Weaknesses:**

1. The problem setting is valid -- being able to robustly control a variety of robot morphologies has direct connections to real world robotic applications (albeit the UNIMAL benchmark has a very limited design space of robots).

2. The motivation for the knowledge diversion formulation is clear and intuitive -- separating task-agnostic learngenes from task-specific tailors is well justified in the proposed cross-morphology, cross-task setting.

3. The authors provide extensive experiments to demonstrate that DivMorph achieves state-of-the-art performance in both sample efficiency and cross-task transfer ability.

Weaknesses:
1. Overly complex terminologies throughout the paper -- what are essentially architecture-level implementation details and matrix refactoring get named various biology-inspired but quite vague terms, which reads a bit forceful.
2. I would recommend replacing all the comic sans font in all the figures.

---

> ### Author Rebuttal · Authors · 2026-03-31
>
> Dear Reviewer pu2W,
>
> We sincerely appreciate your recognition of our motivation and comprehension.
> Below, we provide our detailed response, with experimental tables and figures accessible via anonymous link as permitted by ICML26.
>
> **📎 Anonymous Link**\
> 👉 https://anonymous.4open.science/r/a-D81E/r.pdf
> >**W1: Overly Complex Terminologies**
>
> Following the terminology in Knowledge Diversion[1], we denote weights capturing task-agnostic knowledge as "Learngene", **emphasizing their gene-like transferability**, and those encapsulating task-specific knowledge as "Tailor", **reflecting their adaptability to specific tasks**.
>
> These terms may be more intuitive to Machine Learning community. As suggested, we will instead adopt "task-agnostic weights" and "task-specific weights" to better align with RL conventions.
>
> [1] KIND: Knowledge Integration and Diversion for Training Decomposable Models, ICML'25
> >**W2: Comic Sans Font**
>
> Thank you for the suggestion. We will replace all "Comic Sans" with "Arial" for a more formal presentation.
> >**Q1: Qualitative Behavioral Difference between Different Rewards**
>
> In our experiments, all rewards are continuous—e.g., the agent’s forward speed or distance to a target point or box.
> Unlike discrete rewards in other RL benchmarks, such as a binary success/failure signal or a sparse reward given only upon task completion, **these continuous rewards more faithfully reflect the agent’s task performance**.
>
> In our **Supplementary Material**, we have provided videos of agents performing various tasks to visually illustrate how reward differences manifest in behavioral differences.
> To further quantify the comparison, we present **a frame-by-frame analysis of agents in the Point Navigation task (Re_Fig.10)**, where the reward—on the scale of thousands—reflects the total distance traveled toward the target.
> We compare agents under identical conditions—one achieving a reward of ≈4000 and the other ≈3500. The higher-reward agent reaches the target faster, exhibiting greater forward velocity toward the goal.
> >**Q2: Simulation Engine and Wall-Clock Time**
>
> We use the **MuJoCo simulator**, a widely adopted physics engine in RL for efficient and accurate simulation of articulated bodies and contact dynamics.
> Our work builds on **UNIMAL, an embodied control benchmark**[2] featuring control tasks for over 100 distinct agent morphologies.
>
> Conventional agent control typically **trains a separate MLP for each agent morphology (≈0.5h per agent)**, so training 100 agents independently would take `≈50h of wall-clock time` in total.
> Thus, the Universal Morphology Control task, proposed by MetaMorph[3], aims to **learn a single policy that generalizes across heterogeneous morphologies**, requireing `≈10h of wall-clock time` and serving as our baseline.
>
> MetaMorph reduces redundant training across morphologies, but **each new task still requires training a universal controller from scratch (≈10h per task)**.
> We address this with knowledge diversion, which extracts task-agnostic knowledge from off-the-shelf pretrained models on prior tasks and transfers it to new tasks, substantially reducing training time.
>
> For example, compared with policy transfer baselines such as MetaMorph⚗—which distills knowledge from all pretrained teachers—our method achieves the same performance in only `≈1h of wall-clock time in Exploration`, whereas MetaMorph⚗ requires≈10h.
> Similar speedups are observed in other tasks: `Escape≈1h`, `Point Navigation≈5h`, `Push Box Incline≈4h`, and `Manipulate Box≈3h` to match the ≈10h baseline performance (see Re_Fig.4b).
>
> [2] Embodied Intelligence via Learning and Evolution, Nature Communications\
> [3] MetaMorph: Learning Universal Controllers with Transformers, ICLR
> >**Q3: Role of Text Description**
>
> In knowledge diversion, we aim to disentangle task-agnostic and task-specific knowledge to facilitate zero- or few-shot generalization to new tasks.
> Unlike morphology diversion, where hundreds of morphologies suffice to train a dedicated encoder, task diversion involves only a handful of tasks (five in our experiments), **making it infeasible to train a dedicated task encoder that reliably generalizes to unseen tasks**.
>
> This motivates us to **leverage semantic generalization from language for knowledge transfer**.Accordingly, we provide each task with a text description and encode it using a pretrained text encoder to **obtain a corresponding embedding that serves as its task representation**.
>
> Indeed, without text descriptions, generalizable task embeddings are difficult to obtain.
> In this case, task diversion can **only rely on a 0–1 gating scheme**, assigning each task an independent tailor without cross-task sharing.
> As shown in additional ablation experiments (Re_Fig.5), while this removes the need for a pretrained text encoder and preserves the Learngene’s task-agnostic properties, it **requires randomly initialized Tailors for unseen tasks**, compromising zero- or few-shot generalization.

---

> > ### Author Rebuttal · Reviewer_pu2W · 2026-04-03
> >
> > Thank you to the authors for the detailed follow-up. Indeed the new proposed terminology is more straightforward. Not highest priority but, it would be more helpful if the provided link contains animated videos of the trained policies.

---

> > > ### Author Response · Authors · 2026-04-04
> > >
> > > Dear Reviewer pu2W,
> > >
> > > Thank you for your reply; we are thrilled that our rebuttal successfully resolved your concerns! 😊
> > >
> > > Regarding your suggestion to present the trained policies in video format, while we sincerely wish we could provide a video link, the official ICML guidelines request that authors limit external materials during this phase, explicitly stating that **"links should be used primarily for figures (including tables) and captions that describe the figure (no additional text)."** As a workaround to follow this guidance, we put together the frame-by-frame breakdown in Re_Fig. 10.
> > >
> > > Fortunately, the videos comparing DivMorph and MetaMorph are already included in our **Supplementary Material**, submitted with our original manuscript. We kindly invite you to review those videos, as these videos may help in a more intuitive understanding of performance.
> > >
> > > Best regards,
> > >
> > > Authors

---

### Official Review · Reviewer_ho9M · 2026-03-14

**Soundness:** 2
**Presentation:** 3
**Significance:** 3
**Originality:** 3
**Overall Recommendation:** 4
**Confidence:** 3

**Summary:**

This paper proposes a modular reinforcement learning architecture named DivMorph to address the inefficiency of heterogeneous robot morphology control and cross-task policy transfer. The method utilizes Singular Value Decomposition to factorize the weight matrices of a Transformer controller into basic knowledge units. During training, DivMorph leverages policy distillation from a set of pre-trained teacher policy networks, combined with a dynamic soft gating mechanism conditioned on textual and morphological features, to disentangle knowledge into universal learngenes and task/morphology-specific tailors. When deployed or faced with new tasks, the system prunes irrelevant tailors and fine-tunes the policy using the PPO algorithm. Evaluated on the UNIMAL simulation benchmark, DivMorph achieves a significantly smaller deployment size and higher sample efficiency during fine-tuning.

**Compliance With Llm Reviewing Policy:**

Affirmed.

**Key Questions For Authors:**

1.	True Training Costs: What is the end-to-end total computational cost (e.g., total GPU hours and environment interaction steps) to pre-train all teacher policy networks, execute the knowledge diversion (distillation in Algorithm 1), and perform the subsequent PPO fine-tuning? Please provide an efficiency comparison chart that includes these prerequisite costs to demonstrate the true advantage over training baselines from scratch.
2.	Comparison with General PEFT: Given that DivMorph relies on low-rank factorization and modular recombination, how would it perform if standard PEFT methods (such as LoRA) were directly applied to the Morphology-Aware Transformer for task-specific fine-tuning? What are the concrete advantages of DivMorph's complex mechanism over simpler low-rank adaptation techniques?
3.	Hyperparameter Sensitivity: In Appendix Table 1, the ranks for Learngene and Tailor are set to 96 and 32, respectively. How were these values chosen? How does the rank allocation ratio specifically affect the frequency of Negative Transfer and the final model compression rate?
4.	Sim-to-Real Potential: Does the robustness of the highly pruned and disentangled policy model (retaining only activated tailors) degrade when faced with unmodeled noise in physical environments? Do the authors have any plans or preliminary results for zero-shot or few-shot transfer of this policy to real physical robots?

**Limitations:**

The authors briefly mention the positive impact of their work on scalable policy learning in the Impact Statement section after the conclusion. However, they fail to adequately discuss the limitations of their work. Specifically, the paper omits discussion on the following key constraints:
1.	The heavy reliance on pre-trained multi-task expert models and the resulting massive initial computational overhead.
2.	The potential computational bottlenecks of SVD and orthogonal constraints (Cayley transform) when applied to extremely deep or massive-scale Transformers.
3.	The inherent limitations of pure simulation environments and the lack of analysis regarding the challenges of deployment in the real physical world (Sim-to-Real).
Suggestion: The authors must add a dedicated "Limitations" subsection in the main text to honestly disclose these technical and practical barriers.

**Strengths And Weaknesses:**

Strengths:
1.	Exceptional Deployment Efficiency: Through its disentanglement and pruning mechanisms, the proposed method achieves an impressive 16.7x model compression rate for specific single-agent deployment.
2.	Solid Theoretical Analysis: The authors apply an orthogonality constraint via the Cayley transform when introducing SVD-based knowledge diversion. Appendix A provides rigorous mathematical proofs demonstrating how this constraint prevents anisotropic scaling, thereby theoretically ensuring the stability of gradient propagation during RL training.
3.	Innovative Method Design: DivMorph abandons discrete binary gating in favor of a Mixture-of-Experts (MoE) style Dynamic Soft Gating mechanism that integrates textual semantic instructions and robot topological structures. This provides an excellent inductive bias for zero-shot generalization.
Weaknesses:
1.	Hidden and Substantial Training Costs: The paper heavily promotes its high efficiency in policy deployment and cross-task transfer (e.g., 3.3x sample efficiency improvement) but severely downplays the prerequisite training costs. Appendix B.1 explicitly states that the knowledge diversion phase requires supervised distillation from "a set of teacher policy networks trained on multiple tasks." This means that obtaining DivMorph requires an enormous amount of computational power to train all expert models from scratch beforehand. Excluding this massive pre-training + distillation overhead makes the sample efficiency comparison against baselines fundamentally unfair and highly misleading.
2.	Lack of Crucial PEFT Baselines: DivMorph relies heavily on low-rank factorization (SVD) and recombination of network weights. However, there are already many mature Parameter-Efficient Fine-Tuning (PEFT) methods (e.g., LoRA and its variants) in the literature for large model fine-tuning and multi-task adaptation. The paper only compares against domain-specific baselines like MetaMorph and fails to demonstrate whether its complex SVD + Cayley transform + Soft Gating architecture yields any absolute performance or efficiency advantage over applying standard LoRA directly to a Transformer for multi-task RL fine-tuning.
3.	Arbitrary Hyperparameters and Missing Sensitivity Analysis: According to Table 1 in the Appendix, the rank of the learngene is hardcoded to 96, and the tailor rank to 32. However, there are absolutely no ablation studies regarding these core hyperparameters in the main text. How capacity is allocated between universal knowledge and specific knowledge is critical to the ultimate compression limit and transfer performance. The absence of this analysis weakens the empirical rigor of the paper.
4.	Sim-Only Validation (Detached from Reality): All experiments are conducted exclusively in the simulated MuJoCo environment (UNIMAL benchmark). For a 2026 robot learning paper emphasizing efficient control and deployment, the complete lack of validation or discussion regarding the robustness of this disentangled policy against physical world noise (the Sim-to-Real gap) casts doubt on its practical applicability.

---

> ### Author Rebuttal · Authors · 2026-03-31
>
> Dear Reviewer ho9M,
>
> We sincerely appreciate your recognition of our innovation.
> Below, we provide our detailed response, with experimental tables and figures accessible via anonymous link as permitted by ICML26.
>
> **📎 Anonymous Link**\
> 👉 https://anonymous.4open.science/r/a-D81E/r.pdf
> >**Q1:True Training Cost**
>
> Although training teachers is time-consuming (≈10h per model), **it does not constitute a necessary overhead in our setting**.
> Our work builds on the Universal Morphology Control task and UNIMAL benchmark, allowing us to **leverage publicly available pretrained weights from other methods**.
>
> Moreover, **pretrained models are not intrinsic to our method, as nearly all baselines in our setting leverage them for fine-tuning**.
> For example, MetaMorph⚗ also distills knowledge from all pretrained teachers, while MetaMorph↣ transfers the teacher from the most similar training task for each new task.
>
> Consequently, the **sole additional overhead of our method is the knowledge diversion stage**, with a wall-clock cost of ≈0.5h, as it is performed offline without environment interactions (Re_Tab.1 in link details each component’s overhead).
>
> Importantly, **this overhead (≈0.5h) is negligible compared to the time saved on new tasks**: as shown in Re_Fig.4b, our method reaches MetaMorph⚗’s peak performance with minimal iterations, yielding over 108h of cumulative savings across five new tasks[(9+9+5+6+7)h × 3 seeds].The advantage is even greater compared to MetaMorph trained from scratch (Re_Fig.4c), yielding cumulative savings of over 117h [(9+9+7+8+6)h × 3 seeds].
>
> As suggested, we report end-to-end wall-clock time in Re_Tab.3. For fairness, PPO fine-tuning is measured by the time required to reach the peak performance of training from scratch.
> We also include pretrained model costs(~10h each), **although we consider them unnecessary since they are publicly available and not specific to our approach**.
> >**Q2:Comparison with PEFT**
>
> Although PEFT are not included in paper, **we evaluate two knowledge transfer variants of MetaMorph (MetaMorph↣ and MetaMorph⚗), both using full fine-tuning on new tasks**. Full fine-tuning provides a strong baseline, offering greater adaptation than LoRA when sufficient data and training time are available, and already demonstrates the effectiveness of our method for task-agnostic transfer and task-specific adaptation.
>
> As suggested, we further compare with PEFT in Re_Fig.8 by applying LoRA to a pretrained MetaMorph. With limited trainable parameters, LoRA performs weakly in RL.
> In contrast, DivMorph provides more transferable knowledge and effective adaptation by explicitly disentangling pretrained knowledge into:
> - a task-agnostic Learngene, **enabling stronger cross-task transfer than pretrained MetaMorph**;
> - task-specific Tailors, flexibly composed for new tasks, achieving better few-shot generalization than randomly initialized LoRA.
>
> We further apply LoRA on Learngene(Re_Fig.8), achieving better performance than LoRA on pretrained MetaMorph, highlighting its superior cross-task transfer.
> >**Q3:Hyperparameter Sensitivity**
>
> - For task diversion, the goal is to enhance cross-task transfer. **With a non-zero Tailor rank ($N_T>0$), task-specific knowledge can be partially separated, mitigating negative transfer**(see Re_Fig.1 and Reviewer iJDH(Q3) for the analysis with $N_G=96$ and $N_T=32$).
>
> - For morphology diversion, which targets reduced model size for single-agent deployment. Compared to task diversion, **handling ≈100 morphologies requires stronger morphology-specific representations, thus allocate more Tailor ranks to encourage diversity**.
> In practice, we set $N_G=64$ and $N_T=64$, activating 16 Tailors per agent(TopK=16).
>
>   This configuration balances efficiency and performance: **its cost matches a single-hidden-layer MLP**(see Re_Tab.2, Reviewer kjwa(Q5)) while retaining near-original performance, approaching the budget’s efficiency limit.
>   Further ablations in Re_Fig.2 show only marginal gains at a modestly higher deployment cost, while excessive compression noticeably degrades performance.
> >**Q4:Sim-to-Real**
>
> **We enhance robustness for practical deployment by retaining shared learngenes that capture morphology-agnostic knowledge.**
> Unlike standard pruning, which keeps only morphology-specific parameters, these shared representations help mitigate sim-to-real discrepancies and unmodeled noise.
>
> Real-world deployment is a common limitation in morphology control benchmarks, as **the diversity of agent morphologies (Re_Fig.9) often exceeds what current physical robots—typically standard designs like quadrupeds—can realize**. We will explicitly discuss this in paper.
> >**Limitation**
>
> - Reviewer kjwa(Q4) and Re_Tab.1 show that **the cost of SVD, Cayley and Orthogonal is negligible** with our 5-layer Transformer, which is sufficiently expressive for RL.
> - We will explicitly acknowledge the inherent challenge of real-world deployment in embodied morphology control.

---

### Decision · Program_Chairs · 2026-04-30

**Decision:**

Accept (regular)

**Comment:**

Overall, I find this work technically solid and believe it makes a meaningful contribution to morphology control. The problem it studies is important, and the proposed method is technically interesting. The main initial concerns were about the efficiency claims and the lack of clarity around the training details, but the authors addressed these points well in the rebuttal. I therefore recommend accept.